 

# Hybrid protein assembly-histone modification mechanism for PRC2-based epigenetic switching and memory

Cecilia Lövkvist[1], Pawel Mikulski[2], Svenja Reeck[1,2], Matthew Hartley[1], Caroline Dean[2], Martin Howard[1]*

[1]Computational and Systems Biology, John Innes Centre, Norwich Research Park, United Kingdom; [2]Cell and Developmental Biology, John Innes Centre, Norwich Research Park, United Kingdom

**Abstract** The histone modification H3K27me3 plays a central role in Polycomb-mediated epigenetic silencing. H3K27me3 recruits and allosterically activates Polycomb Repressive Complex 2 (PRC2), which adds this modification to nearby histones, providing a read/write mechanism for inheritance through DNA replication. However, for some PRC2 targets, a purely histone-based system for epigenetic inheritance may be insufficient. We address this issue at the Polycomb target *FLOWERING LOCUS C* (*FLC*) in *Arabidopsis thaliana*, as a narrow nucleation region of only ~three nucleosomes within *FLC* mediates epigenetic state switching and subsequent memory over many cell cycles. To explain the memory's unexpected persistence, we introduce a mathematical model incorporating extra protein memory storage elements with positive feedback that persist at the locus through DNA replication, in addition to histone modifications. Our hybrid model explains many features of epigenetic switching/memory at *FLC* and encapsulates generic mechanisms that may be widely applicable.

*For correspondence:
martin.howard@jic.ac.uk

Competing interests: The authors declare that no competing interests exist.

## Introduction

The mechanistic basis for epigenetic state switching and memory storage is incompletely understood. Advances in our understanding have come from studying the Polycomb Repressive Complex 2 (PRC2) silencing system, which confers stable silencing of genes in organisms ranging from *Drosophila* to *Arabidopsis* to humans (*Holoch and Margueron, 2017*). Key to Polycomb silencing is the deposition of the histone modification H3K27me3 onto genomic targets. The enzymatic complex that adds this modification, PRC2, can recognise existing H3K27me3 and then add more methyl groups to nearby histones in a read/write maintenance mechanism (*Hansen et al., 2008*; *Margueron et al., 2009*). In this way, targets with existing H3K27me3 can generate more H3K27me3 and thereby survive perturbations. This ability is vital at DNA replication where levels of histone modifications, such as H3K27me3, drop by a factor of ~2. This dilution is due to the recycling (*Escobar et al., 2021*) and random partitioning (*Annunziato, 2005*) of parental nucleosomes to daughter DNA strands, but where the inherited nucleosomes remember their spatial position (*Schlissel and Rine, 2019*), at least for repressed chromatin domains (*Escobar et al., 2019*), with repressive H3K27me3 being a genuinely epigenetic mark (*Reinberg and Vales, 2018*). The read/write feedbacks can then fill in the missing H3K27me3 to restore levels similar to those prior to replication. In this paradigm, the histone modifications are the only causative memory elements able to *store* information about silencing through DNA replication. These marks are then *maintained* against losses, including at DNA replication, by the PRC2 enzymatic machinery.

We have been studying the *Arabidopsis* gene *FLOWERING LOCUS C* (*FLC*) as an exemplar PRC2 target to dissect the fundamental mechanisms underlying epigenetic switching and maintenance

(*Finnegan and Dennis, 2007*; *De Lucia et al., 2008*; *Angel et al., 2011*). *FLC* is transcriptionally, and then epigenetically in cis, silenced by environmental exposure to cold temperatures in an all-or-nothing, digital fashion (*Berry et al., 2015*). The epigenetic silencing remains stable for many weeks after plants return to warm conditions, aligning flowering to the favourable conditions of spring. Detailed investigation of the epigenetic silencing has revealed an initial nucleation phase during cold, when the cold-induced protein VERNALIZATION INSENSITIVE 3 (VIN3), together with H3K27me3, are deposited in a specific region, the nucleation region, downstream of the transcription start site at *FLC* (*Yang et al., 2017*). On return to the warm, H3K27me3 then spreads to coat the entire locus. Eventually the nucleation peak is lost with H3K27me3 levels in that region dropping to match the rest of the locus. This resulting 'perpetuated' state is then stable in the *Arabidopsis* standard laboratory wild-type Col*FRI*, but not in an *Arabidopsis* variety Lov-1 from northern Sweden (*Qüesta et al., 2020*). Crucially, studying the nucleated state without spreading is possible through mutants in either the *Arabidopsis* histone methyltransferases, CURLY LEAF (CLF), or the *Arabidopsis* homolog of Heterochromatin Protein 1, LIKE HETEROCHROMATIN PROTEIN 1 (LHP1). In these cases, it is found that the digital silencing memory is not fully stable and instead *FLC* stochastically reactivates in individual cycling cells after a timescale of tens of cell cycles (*Yang et al., 2017*).

Such nucleation and spreading dynamics are a highly conserved feature of many Polycomb systems including in mammalian cells (*Oksuz et al., 2018*). Mutants with impaired ability to 'read' H3K27me3 (*Oksuz et al., 2018*), or where PRC2 is competitively inhibited (e.g. the H3K27M mutation or expressing the H3K27M-mimic EZHIP) (*Jain et al., 2020*), exhibit reduced H3K27me3 levels. Another system with similar dynamics to *FLC* is the mammalian immune gene *Bcl11b*, which also exhibits a stochastic epigenetic switch in cis controlling T-cell fate commitment (*Ng et al., 2018*). In addition, *Drosophila* Polycomb systems contain Polycomb Response Elements (PREs) in their DNA sequence, where H3K27me3 is initially deposited and from where H3K27me3 spreads. This spreading is thought to occur by looping interactions in which DNA sequences distal to the PRE are brought close by (*Ogiyama et al., 2018*). There are, however, differences between the Polycomb system in different organisms. Plants have separate dedicated methyltransferases for H3K27me1 (*Jacob et al., 2009*; *Jiang and Berger, 2017*), although their usage at *FLC* remains unclear. On the other hand, in mouse embryonic stem cells PRC2 is responsible for all H3K27 methylation states (*Højfeldt et al., 2018*). The timescale of H3K27me3 reacquisition after DNA replication appears faster in plants (~2 hr in tobacco cells) (*Jiang and Berger, 2017*) compared to mammalian cells (around 1 day) (*Alabert et al., 2015*). However, these dynamics can be faster in mammalian Polycomb targets with higher H3K27me3 (*Reverón-Gómez et al., 2018*).

Recently, two lines of evidence have emerged suggesting that histone modifications, with their associated read/write dynamics, are not sufficient to explain epigenetic memory storage and maintenance. First, even in cases where the number of nucleosomes involved is small, and therefore vulnerable to loss by dilution at DNA replication, epigenetic memory is still surprisingly persistent. An example is the *lhp1* mutant (*Figure 1A*) where the *FLC* nucleation peak described above was only ~three nucleosomes wide, yet the memory lasted for an anomalously long timescale of tens of cell cycles (*Yang et al., 2017*). This 'metastable' memory persistence was despite an expected $(1/2)^3 = 1/8$ probability of complete parental nucleosome loss every cell cycle at DNA replication. Similarly, experiments on mating-type genes in budding yeast have indicated that epigenetic memory was independent of the number of nucleosomes involved (*Saxton and Rine, 2019*). These experiments demonstrate unexpected memory stability and suggest that histone modifications may not be the only memory elements capable of storing information through DNA replication. Secondly, experiments in *Drosophila* have shown that even in cases with widely distributed H3K27me3, when the PREs are deleted, the memory of silencing collapses over subsequent cell cycles (*Coleman and Struhl, 2017*; *Laprell et al., 2017*). This is not what would be expected if read/write feedbacks were the predominant mode of memory maintenance. Similarly, for *epe1*[+] epigenetically inherited H3K9me in fission yeast, specific DNA sequences are required for memory maintenance (*Wang and Moazed, 2017*). These experiments suggest that nucleation regions or PREs play a privileged role in memory maintenance.

Previously, we developed a model of Polycomb-based epigenetic memory dynamics for the fully spread state in a mammalian context (*Berry et al., 2017a*). The model assumed that PRC2 is recruited to already methylated H3K27me3 histones, where it would then catalyse further addition of H3K27me3 in line with accepted read/write feedbacks. Such silencing was then antagonised by

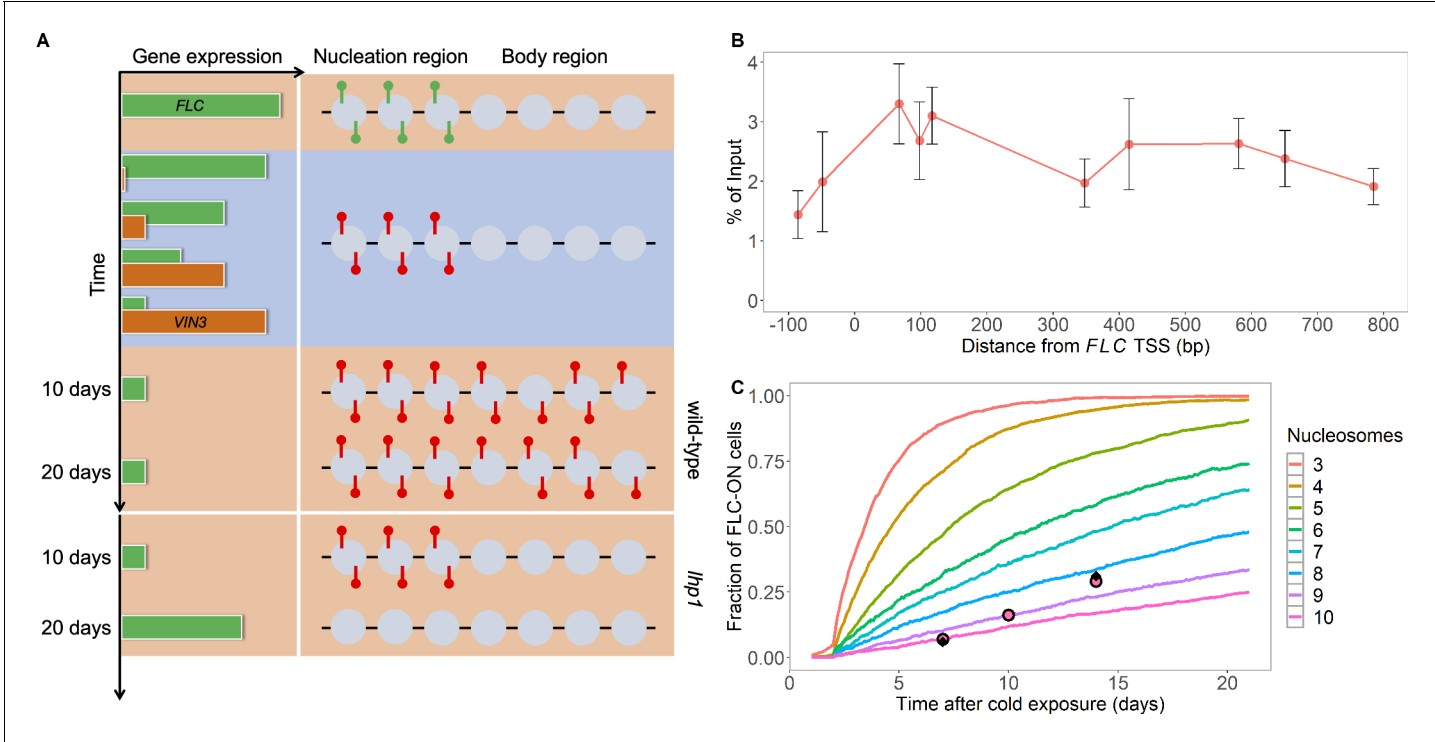

**Figure 1.** Existing model cannot explain persistence of metastable memory at *FLC*. (A) Schematic of *FLC* and *VIN3* expression (left) and chromatin states (right, H3K36me3 modifications in green and H3K27me3 in red) over time, in the warm (red background), in the cold (blue background) and subsequent warm (red background), with wild-type at 10 and 20 days post-cold shown first and *lhp1* mutant at 10 and 20 days post-cold below. (B) MN-ChIP data for H3K27me3 profile (normalised to H3 levels) at *FLC* nucleation region (H3 and H3K27me3 levels with error bars are shown separately in *Figure 1—figure supplement 1*) after a 6-week cold treatment at 5°C. Error bars are sem, from n=3 biological replicates, with separate populations of seedlings harvested from MS plates. (C) Fraction of FLC-ON cells in simulated *lhp1* mutant from model in *Berry et al., 2017a* in warm after cold (full curves, each averaged over 4000 realisations). Number of nucleation-region nucleosomes varied, with assumption that 1/6 of the nucleation region covered in H3K27me2/me3 is sufficient for full silencing, with cells being FLC-ON if this is not satisfied at one or both of the *FLC* copies. Simulations compared to experimental data (black diamonds) from *Yang et al., 2017* and newly acquired data (pink circles) (see also *Figure 1—figure supplements 2*, *3* and *4* and Materials and methods).

The online version of this article includes the following source data and figure supplement(s) for figure 1:

**Source data 1.** MN-ChIP data used in *Figure 1B* and *Figure 1—figure supplement 1*.

**Source data 2.** Fraction of FLC-ON cells used in *Figures 1C* and *2D* and *Figure 1—figure supplement 2*.

**Figure supplement 1.** MN-ChIP experimental results across the *FLC* locus after a 6-week cold treatment at 5°C.

**Figure supplement 2.** Simulated fraction of FLC-ON cells in *lhp1* as a function of time in the warm after cold.

**Figure supplement 3.** Analysis of proportion of FLC-ON cells in *lhp1* after a 10-week cold treatment.

**Figure supplement 4.** Images of *lhp1* mutant roots.

transcription, creating a bistable epigenetic memory system. However, this work did not focus on switching of memory states, nor the persistence of 'metastable' memory states. In this work, we build on this model to create for the first time a full model of epigenetic switching and memory, including the dynamics of nucleation and spreading. Our model is fitted specifically to *FLC* in *Arabidopsis thaliana*, but we emphasise that most of its features are likely to be applicable to any PRC2 target. Our key new prediction is the presence of additional protein memory storage elements with positive feedback, generating assemblies that persist at the locus through DNA replication in addition to histone modifications. We also propose that the formation of assemblies through positive feedback is key for the initial nucleation of the silenced memory state. We further hypothesise that H3K27me3 is spread from the nucleation region to distal regions in the *FLC* gene body through LHP1-mediated looping, permitting long-term memory maintenance. Such a mechanism also rationalises a role for PREs in maintaining epigenetic memory.

The formation of assemblies of proteins is not unprecedented in the Polycomb system, although in somewhat different contexts. Previous work has shown that PRC2 dimerizes, loss of which impairs

H3K27me3 (*Chen et al., 2020*). Dimerization has also been shown in the *FLC* system, for example, in VIN3, but this was again linked solely to histone modification levels (*Greb et al., 2007*). In addition, protein oligomers have been proposed for functioning of the related Polycomb Repressive Complex 1 (PRC1) system: the sterile alpha motif (SAM) domains of Polyhomeotic (Ph), the PhoRC subunit Scm-related protein containing four MBT domains (Sfmbt) and Sex Comb on Midleg (Scm, canonical PRC1 subunit) have been implicated in nucleation, polymerization and Polycomb body clustering (*Isono et al., 2013*; *Frey et al., 2016*). Furthermore, Scm is considered to be an important mediator connecting PRC1, PRC2, and transcriptional silencing, with Scm polymerising to produce broad domains of Polycomb silencing (*Kang et al., 2015*). The PRC1 subunit Posterior Sex Combs (Psc) has also been described as an inherited element through a bridging model (*Lo et al., 2012*). Here, we propose that protein assemblies in the PRC2 system also have a further separate and central role in acting as epigenetic memory storage elements that persist at the locus through DNA replication.

## Results

### Existing model cannot explain metastable memory dynamics at *FLC*

To probe the nucleation region dynamics at *FLC*, we first used our existing model developed for Polycomb silencing generally including mammalian targets (*Berry et al., 2017a*). Briefly, the existing model was constructed around a series of stepwise transitions between low (H3K27me0) and high (H3K27me3) methylated states at each histone. Transitions to more highly methylated states were catalysed through read/write feedbacks by existing nearby H3K27me2/me3. High levels of H3K27me2/me3 silenced transcription, while transcription antagonised H3K27me2/me3, causing methylation loss by, for example, nucleosome eviction (*Schlissel and Rine, 2019*). DNA replication caused loss of pairs of histone modifications on a single nucleosome, with a probability of one half. More details can be found in Materials and methods. We found previously that this model could robustly generate bistable expression states, where either low expression/high H3K27me2/me3 or high expression/low H3K27me2/me3 states are stable (*Berry et al., 2017a*). Note that there are some differences between the mammalian and plant systems, particularly in the rapid addition of H3K27me1 by specialised methyltransferases in plants (*Jiang and Berger, 2017*). However, we found that incorporating a separate, rapid H3K27me0 to me1 transition in the model that was not catalysed by PRC2 did not substantially affect our results, as our existing model already led to rapid H3K27me1 recovery. We therefore initially retained our original model.

We first simulated our original model, but including a specific, small number of nucleosomes representing the nucleation region, which previous work had identified as consisting of about three nucleosomes (*Yang et al., 2017*). To confirm the nucleosome positioning within this region, we performed MN (Micrococcal Nuclease) together with ChIP experiments, MNase-ChIP (*Figure 1B*, *Figure 1—figure supplement 1*). These experiments revealed the presence of two well-positioned nucleosomes with high MNase-ChIP peaks, and then a further less-well positioned nucleosome ~500 bp downstream of the transcription start site, with a broader MNase-ChIP peak, making up the three nucleosome nucleation region. Most other model parameter values were taken from our previous modelling work (*Berry et al., 2017a*), supplemented with previous experimental measurements specific to the *FLC* locus. These latter measurements included the *FLC* transcription rates in the fully active (*Ietswaart et al., 2017*) and fully silenced (*Yang et al., 2014*) states. We also used a cell cycle duration of about one day in the warm, with the cell cycle seven times longer in cold conditions. These and other parameter choices are fully specified and justified in the Materials and methods.

We initially examined the persistence of the metastable memory. Prior experimental data (*Yang et al., 2017*) for the spreading mutant, *lhp1*, suggested a metastable memory timescale of tens of days. We collected additional experimental data in the *lhp1* mutant, measuring the fraction of root cells with reactivated FLC-Venus levels in the warm after 10 weeks cold treatment. Our results, with higher time resolution in the warm (7 days, 10 days, 14 days) than in *Yang et al., 2017* confirmed the long reactivation timescale (*Figure 1C*), with only ~30% reactivation after 14 days in the warm. We next simulated the metastable memory dynamics in our above model, conservatively assuming that only one sixth of the nucleation region histones in the H3K27me2/me3 configuration are sufficient for full silencing. We assumed here and later that reactivation of only one of the two (independent) copies of *FLC* in a simulated cell would be sufficient for overall cellular reactivation,

with reactivation occurring if the fraction of H3K27me2/me3 dropped below the fully silenced threshold. We started from an initial simulation configuration with three fully methylated nucleosomes in the nucleation region, and all other nucleosomes unmethylated, at the start of an initial post-cold time lag of one day before cell division resumes, and incorporating a further lag of 1 day for reactivation to generate visible protein (*Yang et al., 2017*) (Materials and methods). To match the observed lack of spreading in the *lhp1* mutant, we set the PRC2-mediated methylation rate outside the nucleation region to zero to prevent non-nucleation region methylation. With these rules, we found that the simulated metastable memory did not last nearly as long as in our experiments (*Figure 1C*), with ~50% reactivation occurring in much less than 5 days. If one third of the nucleation region histones in the H3K27me2/me3 configuration are needed for full silencing, reactivation is even more rapid (*Figure 1—figure supplement 2A*). Even if we allow our experimental quantification of the FLC-Venus signal to be much less strict (with fewer off cells, *Figure 1—figure supplement 3* and Materials and methods), the resulting experimental metastable memory duration is still too long to agree with our simulation results. This simulation analysis improved on previous considerations by simulating the full model with three nucleation region nucleosomes rather an idealised 'best case' scenario for memory maintenance (*Yang et al., 2017*). Indeed, the new simulations showed that the three nucleation-region nucleosomes were even further away from providing adequate memory maintenance than previously reported (*Yang et al., 2017*). We found that 8–10 nucleation region nucleosomes were required to match the experimental metastable memory duration (*Figure 1C*), a number which is much too large. Even in this case, the shape of the simulated reactivation profile did not match that found experimentally (*Figure 1C*). In essence, because of the small number of memory-carrying nucleosomes, the probability that the memory of silencing is lost at DNA replication due to unequal partitioning between the two daughter strands is too large, leading to rapid transcriptional reactivation. The existing model cannot therefore explain how the three-nucleosome nucleation region can metastably store the memory of silencing for tens of cell cycles (*Berry et al., 2017a*; *Yang et al., 2017*). Taken together, these results show that the existing model must be amended to handle metastable memory maintenance at *FLC*.

## Generalised epigenetic memory model with additional protein memory storage elements

In order to explain the persistence of the *FLC* metastable memory observed above, there are two possibilities: either the known memory elements are more stable than previously thought, or there are additional memory elements acting to boost the stability of the memory. For the first possibility, the partitioning of the H3K27me3 memory elements at DNA replication would need to be more ordered, and not entirely random as previously assumed. This alternative is at least in principle feasible, due to the existence of specific factors at DNA replication forks, promoting either leading or lagging strand deposition (*Gan et al., 2018*; *Petryk et al., 2018*; *Yu et al., 2018*), and by evidence for regulated biased segregation in *Drosophila* germline stem cells (*Wooten et al., 2019*). For the second possibility, there would need to be additional memory storage elements to H3K27me2/me3. An optimal example of the first possibility would be a strictly alternating distribution of successive nucleosomes from the parental to the two daughter strands, so that at *FLC* each daughter strand would receive at least one of the three parental nucleosomes, thus more stably preserving the memory. The ordered distribution hypothesis does, however, conflict with previous data on nucleosome dynamics at DNA replication (*Annunziato, 2005*). These data were not collected specifically for Polycomb targets, so it remains formally a possibility that at appropriate Polycomb-target genes the nucleosome distribution after DNA replication could be more ordered. However, as yet we know of no evidence specifically pointing in this direction. For completeness, we analyse this alternative model in the Materials and methods (*Figure 1—figure supplement 2B*). Even in the best-case alternating distribution case, we find that more than three nucleation region nucleosomes are needed (*Figure 1—figure supplement 2B*), more than observed experimentally, and again the shape of the simulated reactivation profile does not match that found experimentally. For these reasons, we do not favour models based solely on ordered nucleosome partitioning.

The second possibility is that there are extra, non-histone modification memory storage elements in the *FLC* nucleation region. These elements could stabilise the metastable memory due to their greater number, which would reduce the probability that they will be so unevenly partitioned at DNA replication that memory on one daughter strand would be lost. For these extra elements to

contribute, they cannot be other histone modifications, as at DNA replication if a nucleosome is distributed to the other daughter strand then all histone marks are lost.

A further conceptual problem concerns how histone modifications can first be deposited during nucleation in the cold, as the read/write mechanism requires H3K27me2/me3 to already be present before feedbacks can add more. A low level of H3K27me3 can be generated by noisy, background processes, but these could be insufficient to effectively nucleate a H3K27me3 peak in the cold. Accordingly, we hypothesise that the new elements are also needed to initially create a H3K27me3 nucleation peak in the cold and must therefore be deposited in the nucleation region before H3K27me3.

To satisfy these stringent constraints, we propose a protein assembly forming at the nucleation region, as we detail below. Clear candidates that could make up such an assembly are PRC2-interacting proteins, potentially the VIN3, VERNALIZATION 5 (VRN5) and VEL1 proteins that immunoprecipitate with PRC2 (*Wood et al., 2006*; *De Lucia et al., 2008*). These proteins contain PHD fingers proposed to facilitate recruitment at *FLC* (*Kim and Sung, 2013*) (see below) and a protein interaction (VEL) domain at their C-terminus, which facilitates homo and heterodimerization in yeast and plant cells, which was linked to histone modification levels (*Greb et al., 2007*). VIN3 expression is upregulated by cold (*Sung and Amasino, 2004*), and both VIN3 and VRN5 proteins associate with the nucleation region at *FLC* (*Sung and Amasino, 2004*; *Yang et al., 2017*). Longer cold leads to higher VIN3 levels and hence a potentially higher local concentration, leading to establishment of the H3K27me3 nucleation peak at that locus. In the following, we develop a model based on VIN3/VRN5, although we emphasise that other proteins may also be involved.

Polycomb proteins are recruited to their targets (here, the nucleation region) through a variety of mechanisms: sequence-specificity, transcription factors, lack of CG methylation, histone modifications and DNA structure (*Skourti-Stathaki et al., 2019*; *Yu et al., 2019*). We propose that a combination of these factors brings the putative assembly subunits to the nucleation region. Here, for example, the upstream factor VIVIPAROUS1/ABI3-LIKE 1 (VAL1) can bind to the nucleation region due to sequence specificity and indirectly interact with VIN3 (*Qüesta et al., 2016*). When a sufficiently large number of VIN3 proteins (here, four) are present by chance simultaneously, a nucleation threshold is overcome, and a metastable, dynamically exchanging assembly forms. The assembly can then attract more proteins through strong positive feedback, counterbalancing monomer loss in a dynamic steady state. Hence, the assembly is a digital phenomenon: either present or not present at any given locus, consistent with the properties of metastable silencing in individual cells (*Angel et al., 2015*; *Yang et al., 2017*). Once formed, the assembly stimulates catalysis of H3K27 methylation through stimulation and/or recruitment of PRC2 factors (*Figure 2A*), thereby forming and then maintaining the H3K27me3 nucleation peak.

The act of nucleation thus involves formation of the assembly and subsequent addition of H3K27me3. If the nucleation-region histone modifications are lost at DNA replication and replaced by newly incorporated, unmarked nucleosomes, then surviving members of the assembly can recreate the H3K27me3 peak. We assume here that components of the assembly persist through DNA replication at the locus and are also randomly distributed to the daughter strands at DNA replication similar to, but separate from, the random nucleosome redistribution. Less random segregation might also be possible (as we consider explicitly for the nucleosomes). However, an equal distribution to the two daughter strands is less likely as this could generate memory that is too persistent. In this way, assembly components also constitute epigenetic memory storage elements, conceptually similar to histone modifications. Passage of the replication fork will certainly disrupt the assembly, but if members of the assembly can be transiently retained around the replication fork, rather than diffusing away, then seeding of two daughter assemblies on the two daughter strands could be efficient. For this reseeding process to be effective, we note that the assembly size before replication must be more than twice the minimal size needed for a stable assembly, due to an approximate two-fold dilution at DNA replication. In our fitted model below this constraint is satisfied, as the assembly size (around 17) is much bigger than twice the minimum assembly size (fixed at 4). Only in rare cases where enough assembly members are lost at DNA replication, so that the assembly copy number drops close to or below the nucleation threshold, will nucleation sometimes be lost. Note that members of the assembly feedback on themselves and onto nucleation region H3K27me3 (on all nucleation region histones equally), but we assume that H3K27me3 does not feedback to recruit extra members of the assembly. The rationale for this choice will be explained later. We also allow for

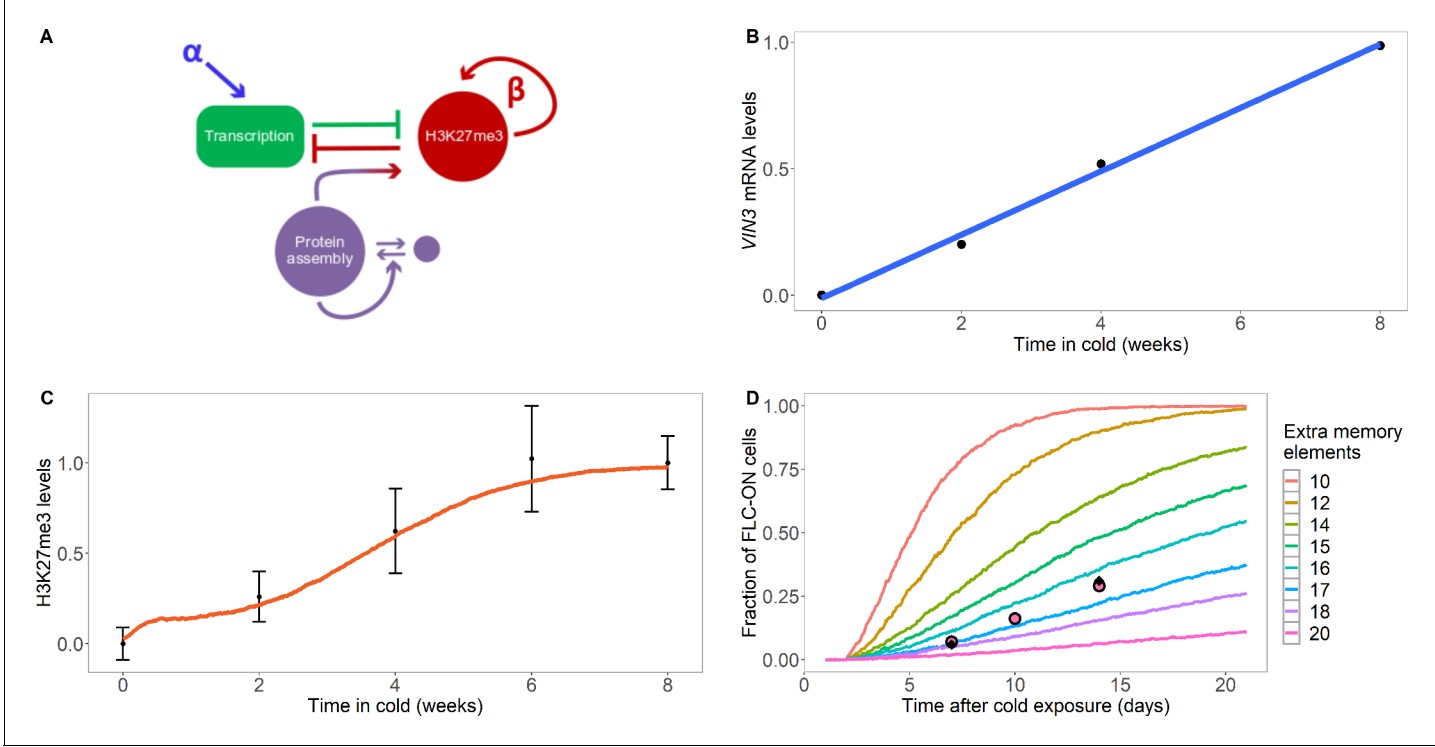

**Figure 2.** Model with extra assembly memory elements can explain nucleation dynamics and metastable memory at *FLC*. (**A**) Schematic of the model with protein assembly as extra memory elements (see *Figure 2—figure supplement 1* for a detailed schematic of the model). (**B**) *VIN3* mRNA levels (black circles) in 8°C cold treatment for Col*FRI*, replotted data from *Hepworth et al., 2018*, blue line: linear fit. (**C**) Simulated H3K27me3 nucleation peak dynamics in cold (fraction of maximum possible occupancy; red line, averaged over 4000 realisations), with extra protein assembly memory elements, compared to data from *Yang et al., 2014* (black circles, error bars: sem) for wild-type Col*FRI*. (**D**) Fraction of FLC-ON cells (full curves, each averaged over 4000 realisations) in simulated *lhp1* mutant using model with varying number of non-histone modification memory elements, in addition to modifications on three nucleosomes, and with the assumption that 1/3 of the nucleation region covered in H3K27me2/me3 is sufficient for full silencing, with cells being FLC-ON if this is not satisfied at one or both of the *FLC* copies. Comparison shown to experimental data (black diamonds) from *Yang et al., 2017*, and newly acquired data (pink circles) (see also *Figure 1—figure supplement 3* and Materials and methods). Details of the model, simulations and parameters are found in Materials and methods and *Figure 2—figure supplement 2*.

The online version of this article includes the following figure supplement(s) for figure 2:

**Figure supplement 1.** Detailed schematic of feedbacks in mathematical model including assembly dynamics.

**Figure supplement 2.** Varying $p_{dem}$ and $p_{ex}$ in the histone-feedback only model to find a bistable region.

H3K27me3 to feedback to add more H3K27me3 on nearest-neighbour histones in the nucleation region in an allosteric read/write mechanism, but at lower rates. This could occur through PRC2 being constitutively recruited close to the nucleation region independently of H3K27me3 consistent with the methyltransferase CLF being detectable by ChIP in that region before cold (*Yang et al., 2017*) and then being allosterically activated by nearby H3K27me3 (Materials and methods).

We next used existing data on the rise of VIN3 protein levels during cold to help parameterise our model of nucleation region dynamics, which showed that VIN3 levels rise linearly with time in the cold (see *Figure 2B* and *Hepworth et al., 2018*). We also allowed for a reduction in the push to transcribe in the cold to represent the non-epigenetic antisense ('*COOLAIR*') pathway which downregulates *FLC* expression (Materials and methods). This effect encourages H3K27me3 nucleation by reducing the antagonising effects of transcription (*Berry et al., 2017a*). We also assumed a short lifetime for VIN3 in the nucleation region, with a timescale of minutes, qualitatively motivated by the rapid recovery of PRC2 subunits previously observed by FRAP (*Youmans et al., 2018*). Hence, the nucleation region assembly, once formed, is a highly dynamic object. Early in the cold, VIN3 levels are low, which leads to a low probability that sufficient VIN3 proteins come together to initiate the assembly; naturally explaining the low nucleation probability at that time. Interestingly, we found that later in the cold, we could only fit the further rise of the H3K27me3 nucleation peak by assuming

that eventually VIN3 is no longer limiting for nucleation and that the probability of nucleation saturates at a high enough VIN3 concentration. Without this assumption, it was not possible to fit the sigmoidal form of the population-level H3K27me3 nucleation peak rise. The model therefore predicts that other components are also needed for nucleation that eventually become limiting after a long enough duration of cold. The nucleation region model (see *Figure 2A*, *Figure 2—figure supplement 1* and Materials and methods for a full description) could then explain the population-level rise in H3K27me3 levels in the nucleation region during the cold (*Figure 2C*): as time passed in the cold, more and more loci digitally nucleate the assembly helped by rising VIN3 levels, leading to a digital H3K27me3 nucleation peak, and therefore a smoothly rising H3K27me3 peak at a cell population level. In these simulations, we used a reduced value of the PRC2-mediated methylation rate $k_{me}^{low}$ outside the nucleation region to inhibit non-nucleation region methylation in the cold, as observed experimentally.

We could also now explain the persistence of the metastable memory in the warm following cold in the *lhp1* mutant. During this phase, no further de novo nucleation was permitted in the model, consistent with VIN3 being depleted in experiments within hours of reintroduction to warm temperatures (see below) (*Hepworth et al., 2018*). We also set the PRC2-mediated methylation rate outside the nucleation region to zero to prevent non-nucleation region methylation and initialised the simulation with three fully H3K27me3 occupied nucleation region nucleosomes, as we did previously for *Figure 1C*, as well as the maximum number of assembly members (Materials and methods). We also assumed that one third of the nucleation region histones in the H3K27me2/me3 configuration are sufficient for full silencing. With the extra non-histone modification memory elements, we found that the nucleation region memory was preserved for long enough to match our experimental data (*Figure 2D*), with the best fit requiring 17 extra memory elements in the assembly. The fit of the simulated reactivation profile to the data was also much better than that found previously in *Figure 1C*, with slower initial reactivation which then accelerates with time in the warm. This shape is due to the need to first lose the assembly before H3K27me3 can then be lost; such a two-step process leads to a sigmoidal reactivation profile. In addition, the model was also able to explain the low levels of H3K27me3 experimentally observed in the nucleation region prior to cold (and arrival of VIN3): the initially low levels of H3K27me3 were unable to generate enough allosteric activation through constitutive PRC2 activity to allow these levels to increase. Note that the fit in *Figure 2D* while good is not perfect, indicating that our model may still be too simple. One possible extension is a combined model with both ordered nucleosome distribution and extra memory elements. This generalised model can also explain our data, using a strictly alternating nucleosome distribution at DNA replication and a best fit reduced number of eight memory elements (*Figure 1—figure supplement 2C*). However, we emphasise that the extra memory elements are still required.

One potential difficulty of the extra memory element model for *FLC* is that the VIN3 protein disappears rapidly after the end of cold treatment, yet the metastable memory is persistent for many days in experiments in the subsequent warm, requiring that the assembly memory component also persists in the warm. Hence, other factors must be involved, potentially the constitutively expressed VIN3 homologues, VRN5 and VEL1 or components of a PRC1 complex. For simplicity in the model, we model a single protein unit capable of positive feedback once a critical local concentration has been reached at the nucleation region. However, consistent with nucleation only occurring during the cold, spontaneous binding to the nucleation region is only allowed in the model during the cold, but with positive feedback possible at all times both in the cold and warm. With this reasoning, the hybrid assembly/H3K27me3 model can explain all our data on nucleation and persistence of metastable memory.

## Looping model for spreading dynamics

So far, we have dissected the dynamics of the nucleation region, in the cold and then in the warm in the *lhp1* mutant. However, in the wild type, the H3K27me3 domain expands from the nucleation region to cover the entire locus after transfer to warm conditions, in a process we term spreading from the nucleation region to the body region. Up to now, we have limited this spread, as appropriate for the wild-type in the cold and *lhp1* mutant in the warm. We now relax this restriction for the wild-type, using a value $k_{me}^{high}$ of the PRC2-mediated methylation rate outside the nucleation region in the warm, and examine in more detail how this spreading process occurs.

Interestingly, at all times when present, both in the cold and in the subsequent warm, VIN3, VRN5 and the PRC2 methyltransferases CLF and SWINGER (SWN) are predominantly located in the nucleation region (*Yang et al., 2017*). There is some limited spreading of these factors but only to significantly lower levels. Although alternatives such as nucleosome mobility have previously been proposed (*Chory et al., 2019*), we consider here the possibility that spreading is mediated by a looping interaction between the nucleation region-located methyltransferases and nucleosomes in the rest of the *FLC* gene body. This interaction would be allosterically activated by H3K27me2/me3 on the target nucleosome. During this interaction, the key proteins maintain their presence in the nucleation region consistent with previous protein ChIP data. Such a mechanism is also consistent with PRC2 being present at unexpectedly low levels, even in cases with active cell cycling, with only one methyltransferase for every ~20 H3K27me3 marks in *Drosophila* (*Bonnet et al., 2019*). Localising the methyltransferases in discrete regions, and then allowing for a longer-range looping interaction to a larger domain, can clearly make efficient use of limited number of methyltransferases to reliably coat extended domains with H3K27me3 (*Oksuz et al., 2018*).

We therefore assumed the constitutively present methyltransferases CLF and SWN close to the nucleation region were able to loop to contact with other body-region nucleosomes at the *FLC* locus and, if H3K27me2/me3 was present on the body region nucleosome, then more H3K27me could be allosterically added to that nucleosome or its nearest neighbours. Although, in this interpretation, the addition of H3K27me is meditated by looping from the nucleation region, mathematically this process turns out to be identical to the local spreading process implemented in *Berry et al., 2017a*. This is because H3K27me2/me3 must be present on the target body nucleosome, before more H3K27me can be added to that nucleosome or its neighbours. Hence, effectively it is this neighbouring H3K27me2/me3 that catalyses the addition, with the looping interaction simply enabling the reaction to proceed. Note that we assume here, for simplicity, that such looping could occur anywhere within the locus with equal probability (see below), but not outside, and that this probability was much higher in the warm than in the cold in the wild-type. Such looping could be mediated, at least in part, by the formation of discrete LHP1 foci, as mutations in *lhp1* (in the RNA binding domain) that compromise the foci also compromise efficient spreading (*Berry et al., 2017b*). Within the LHP1 foci, we suppose that looping interactions are relatively unconstrained, facilitating the spreading process, but that these interactions are much reduced outside. We emphasise that the looping interaction and addition of H3K27me in the body is not directly enhanced by the assembly: the assembly's only effect on histones is to enhance H3K27me3 in the nucleation region. Hence, with the exception of the assembly dynamics itself, and its role in enhancing nucleation region H3K27me2/me3, the underlying model is similar to that of *Berry et al., 2017b*.

With these considerations, we then simulated the full nucleation and spreading dynamics in the Col*FRI* wild-type (*Figure 3A*). Beginning in the warm, the initially low level of H3K27me3 ensured limited allosteric activation and therefore continued low levels of H3K27me3 across the whole locus. In the cold, nucleation occurred, as discussed above. With a low level of looping in the cold, low levels of spreading across the locus could be generated, as observed experimentally. On return to simulated warm, looping was then much more extensive, and it was then possible to generate H3K27me3 spreading dynamics in the warm similar to previously published ChIP data. Note that we do not explicitly fit to warm ChIP data, because the spreading dynamics at a whole plant level (as assayed by ChIP) is complex, as only cells with an active cell cycle can spread (*Yang et al., 2017*). At subsequent DNA replication events, any dilution of H3K27me3 in the gene body was filled in by the looping interaction/allosteric activation before the next round of replication (*Figure 3A,B*). In parameterising these processes, we considered prior data from tobacco cells where H3K27me3 dilution at DNA replication was reversed after only ~2 hr (*Jiang and Berger, 2017*), much faster than in mammalian systems (*Alabert et al., 2015*). For our *Arabidopsis FLC* simulations, our best fit used a timescale intermediate between these cases. Slower rates resembling mammalian systems were unable to generate effective spreading, with limited spreading diluted away by DNA replication, contrary to experiments.

## Loss of the assembly state in the wildtype

Our previous ChIP results in the warm demonstrate that even in a fully spread state in the standard laboratory wild-type Col*FRI*, the population-average H3K27me3 levels in the nucleation region drop to those of the body (*Yang et al., 2017*; *Qüesta et al., 2020*). We previously referred to this new

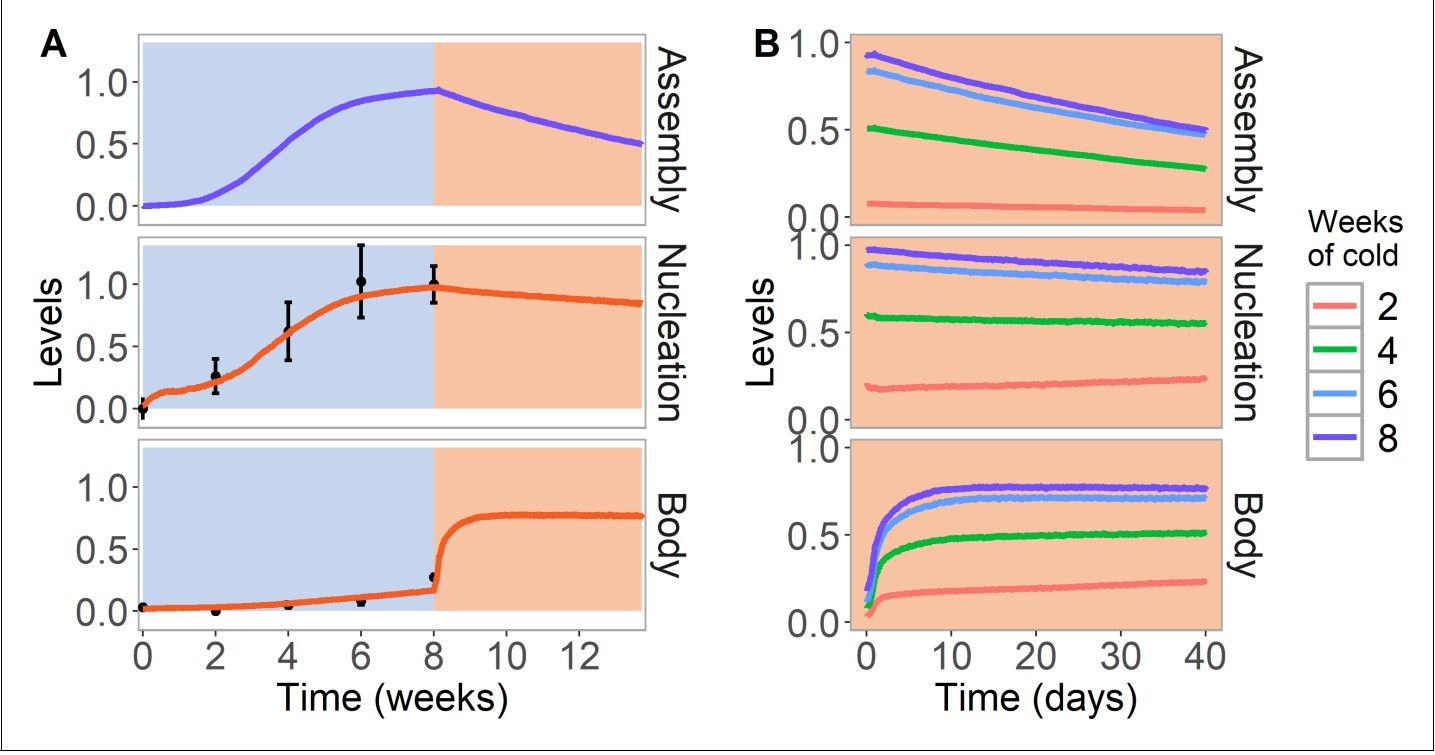

**Figure 3.** Simulation of cold treatment followed by warm conditions in model with assembly positive feedback, including both nucleation region and gene body. Simulated dynamics (lines) for assembly, as well as nucleation and body region H3K27me3 levels, for wild-type Col*FRI*. (**A**) Cold treatment (blue background) is followed by warm conditions (red background). Simulated H3K27me3 nucleation levels in cold same as *Figure 2C*. Experimental H3K27me3 ChIP data (black circles, error bars: sem) (*Yang et al., 2014*). (**B**) Simulated dynamics in the warm, after 2, 4, 6, or 8 weeks of cold treatment. The 8 week data in the warm is the same as in (**A**). Simulated assembly and H3K27me3 levels are fraction of maximum possible occupancy in relevant region, each averaged over 4000 realisations.

state, with lowered nucleation region H3K27me3, as the 'perpetuated' state (*Qüesta et al., 2020*), as in Col*FRI*, this state is stable.

These results are straightforward to understand within our model. As mentioned previously, assembly positive feedback assists in the addition of H3K27me but not vice versa. Hence, eventually, the assembly will suffer stochastic subunit loss at DNA replication to below the nucleation threshold. Due to the lack of feedback from H3K27me, the remaining subunits will then disperse (*Figure 3A,B*, top). This then leads to a reduction in the H3K27me2/me3 feedbacks and levels in the nucleation region (as observed experimentally *Yang et al., 2017*; *Qüesta et al., 2020*) potentially due to the lack of assembly proteins to boost PRC2 activity (*Figure 3A,B*, middle). Note that this reduction is only experimentally observed to occur in the nucleation region not elsewhere: hence, the model was constructed such that the assembly only positively feedbacks onto H3K27me3 in the nucleation region and not elsewhere, as mentioned previously. The methyltransferases CLF and SWN are, however, still present in the nucleation region, due to constitutive binding, as revealed by ChIP data (*Yang et al., 2017*). Therefore, the addition of H3K27me across the locus through the looping interaction is still present. Due to the higher levels of H3K27me2/me3 across the locus, we found that CLF/SWN activity from the nucleation region can still perpetuate the silent state using long-distance looping/allosteric activation, even without assembly proteins replenishing the nucleation region H3K27me2/me3. Such feedback is not present at such high levels during warm periods prior to cold exposure due to the lack of sufficient H3K27me2/me3 to permit significant allosteric activation, as discussed previously. This inherent bistability ensures that there is no premature rise in H3K27me3 levels before cold.

## Nucleation region as a Polycomb Response Element

There are many parallels between the *FLC* nucleation region and Polycomb Response Elements (PREs), as both are required in their respective contexts for epigenetic state switching. We therefore asked if further insight could be gained by treating the *FLC* nucleation region as an effective PRE. If so, our model should capture the results of PRE (nucleation region) excision experiments, where epigenetic memory is lost after 1–6 further cell divisions after excision (*Coleman and Struhl, 2017*; *Laprell et al., 2017*). Indeed, we recover very similar results in our model (*Figure 4A*). The nucleation region, with its constitutively bound CLF/SWN, and its looping interaction, are essential for maintenance of the silenced state in the spread and perpetuated states. As soon as the PRE nucleation region (with any assembly elements) is removed, the feedbacks are greatly diminished (*Figure 4A*, red, yellow curves) and the silenced state decays over the subsequent few cell cycles. In this case, the decay of the silencing is somewhat slower than would be predicted just from nucleosome dilution at DNA replication (*Figure 4A*, pink curve) as we allow here for the effect of limited PRC2 binding throughout the gene body which is able, at a low level, to add further H3K27me through standard read/write feedbacks. This scenario has also been reported experimentally (*Coleman and Struhl, 2017*). However, this feedback mechanism can easily be adjusted to be too weak to perpetuate the state fully, in agreement with PRE excision experiments.

## Loss of the perpetuated state in natural variants

To explore the model further we also simulated the *Arabidopsis* variety Lov-1 from northern Sweden, that reactivates *FLC* expression most markedly if it is not exposed to sufficiently long cold (*Qüesta et al., 2020*), in contrast to the standard laboratory wild-type Col*FRI*. In previous work, we traced this reactivation to an instability in the perpetuated state. One possible explanation for this reactivation would be that *FLC* in Lov-1 has an intrinsically higher transcriptional activity in the warm, with the higher antagonistic activity eventually overwhelming the H3K27me3-mediated silencing. In support of this idea, Lov-1 differs in cis from Col*FRI* at *FLC* through Single Nucleotide Polymorphisms (SNPs) specifically in the nucleation region, SNPs that could alter trans factor recruitment strengths. To further test this hypothesis, we therefore simulated stronger transcriptional activation (α) in warm conditions (*Figure 4B,C*) for Lov-1, as well as slightly altered nucleation region parameters in the cold (Materials and methods), again consistent with nucleation region SNPs. The stronger transcriptional activation, together with the shorter cell cycle in the warm versus cold, caused a loss of silencing at the population level in the warm. The eventual H3K27me3 loss occurred through H3K27me3 levels dropping too low at a given locus after DNA replication to be recoverable in the subsequent cell cycle, due to the stronger antagonistic effect of transcription (*Figure 4C*). Consistently, in simulated non-replicating Lov-1 cells (*Figure 4—figure supplement 1*), the silencing was stable. However, this reactivation could only occur after loss of the assembly, as otherwise the assembly memory elements could simply re-nucleate H3K27me3. Accordingly, we found there was a delay before transcriptional activation could occur, as proposed in *Qüesta et al., 2020*. In this way, through subtly impaired bistability due to increased transcriptional activation, the model was able to give a simple explanation for the eventual reactivation behaviour observed in replicating cells in Lov-1, but not in *Col*FRI.

## Discussion

In this paper, we have developed a full model of epigenetic switching and memory including nucleation and spreading at *Arabidopsis thaliana FLC*, significantly extending the conventional H3K27me3 memory element and read/write feedback paradigm. Our model provides a comprehensive description of the silencing process from nucleation to spreading to the perpetuated, silenced state and in some cases to reactivation. Model output is in good agreement with our data in all cases, showing that our model is sufficient to explain the data. The model also makes several clear predictions: most notably, that additional memory storage elements are required for PRC2-based memory and these involve proteins clustered in an assembly that persist through DNA replication at the locus. Possible proteins that might make up these assemblies at *FLC* are VIN3 and VRN5, although other proteins are also plausible candidates (e.g. VEL1). In the future, it will be very interesting to image the spatiotemporal dynamics of these proteins to examine whether assemblies exist and whether

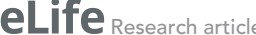

**Figure 4.** PRE excision simulation and simulated reactivation in Lov-1 accession. (**A**) H3K27me3 levels over the entire *FLC* locus in a simulated PRE excision experiment. A fully spread state is simulated; the nucleation region (with any assembly elements) is removed (at day 11 in the simulations), thereby interrupting the looping reactions and compromising the spread state. Varying levels of PRC2 feedbacks ($k_{me}^{ex}$, right) are simulated in the body region when the nucleation region is removed, with each level averaged over 4000 realisations. (**B,C**) Simulated dynamics (lines) of assembly, as well as nucleation and body region H3K27me3 levels, for Lov-1 accession. Lov-1 is simulated with the parameter for transcription activation ($\alpha$) in the warm higher than in Col*FRI* (specified in Materials and methods). (**B**) Cold treatment (blue background) is followed by warm conditions (red background). Experimental H3K27me3 ChIP data (black circles, error bars: sem) (*Qüesta et al., 2020*). (**C**) Simulated dynamics in the warm after 2, 4, 8, or 12 weeks of cold treatment. The 12-week data in the warm is the same as in (**B**). Simulated assembly and H3K27me3 levels are fraction of maximum possible occupancy in relevant region, each averaged over 4000 realisations. In *Figure 4—figure supplement 1*, non-replicating Lov-1 cells are simulated. The online version of this article includes the following figure supplement(s) for figure 4:

*Figure 4 continued on next page*

*Figure 4 continued*

**Figure supplement 1.** H3K27me3 levels are stable after cold in Lov-1 in simulated non-replicating cells.

they colocalise to the *FLC* gene. We also propose that positive feedback dynamics are key for the initial nucleation of the assembly and of the silenced memory state. We further hypothesise that the spreading process following nucleation is a long-range looping process mediated by a separate LHP1 and RNA-dependent focus. Such a mechanism is compatible with a need for Polycomb Response Elements to retain epigenetic memory. It has also recently been suggested that chromatin compaction itself may form part of a memory system (*Pease et al., 2021*). While we cannot definitively rule out this possibility, or the presence of alternative extra memory element systems, the small size of the nucleation region at *FLC* argues against this specific possibility in this case.

Summarising our model (*Figure 5*), we propose that the first step in silencing is stochastic nucleation of proteins in a local assembly. This assembly then permits the de novo addition of nucleation region H3K27me3, with both assembly components and histone modifications acting as memory storage elements. After return to the warm, spreading of H3K27me3 across the locus is mediated by a long-range looping interaction whose range may be controlled by the formation of larger scale, LHP1-RNA clusters. Eventually, the assembly memory storage elements at the locus fall apart after stochastic subunit loss at DNA replication. Following this event, the resulting perpetuated state, which relies purely on H3K27me3 memory elements, is stable in some accessions; however, in others it eventually decays. This decay could be due to increased transcriptional activation, leading to a defect in rebuilding the silenced state after DNA replication.

Although our model has been constructed for *Arabidopsis FLC*, its elements are likely conserved across many systems, including in mammals and flies. Indeed, our model is based on an analogy between the nucleation region and PREs in *Drosophila*. Our model does not yet include PRC1 and H2A ubiquitination, which have also been implicated in Polycomb silencing (*Blackledge et al., 2020*; *Moussa et al., 2019*). It will also be interesting to explicitly include in our model activating histone marks such as H3K36me3. A further extension could be to include nascent transcripts titrating away PRC2 components due to competitive RNA binding (*Ringrose, 2017*; *Wang et al., 2017*), thereby adding further antagonism between transcription and PRC2-mediated silencing. Our model may also rationalise results from *Højfeldt et al., 2018*, suggesting that in mouse embryonic stem cells, H3K27me3 patterns can be accurately established de novo in cells that have lost all H3K27 methylation. In our model, this could in part be due to assembly proteins continuing to mark regions normally occupied by H3K27me and facilitating H3K27me3 renucleation.

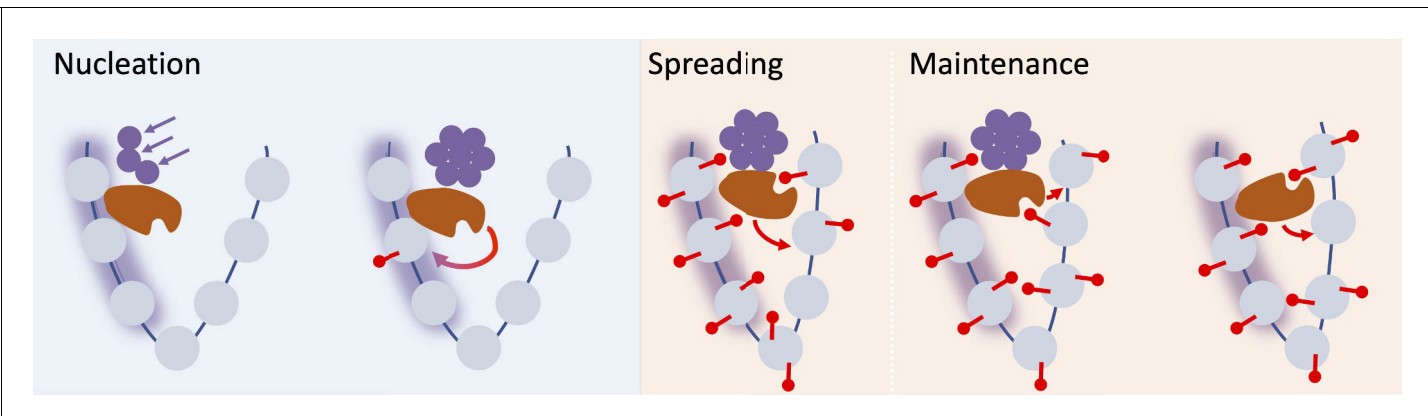

**Figure 5.** Schematic of nucleation, spreading, and maintenance at *FLC*. First, stochastic nucleation of an assembly occurs in the cold and permits the de-novo addition of nucleation region H3K27me3. On return to warm, a long-range looping interaction mediates spreading of H3K27me3 across the locus. Eventually, the assembly memory storage elements at the locus are lost, but H3K27me3 is in most cases maintained at the locus, but can eventually be lost if the transcriptional push is high enough (e.g. in Lov-1 variety).

# Materials and methods

## Key resources table

| Reagent type (species) or resource | Designation | Source or reference | Identifiers | Additional information |
|---|---|---|---|---|
| Gene (*Arabidopsis thaliana*) | *FLC* | TAIR | AT5G10140 | |
| Strain, strain background (*Arabidopsis thaliana*, Columbia-0) | 'Col-*FRI*; Col-*FRI*[sf2]' | *Yang et al., 2017* DOI:10.1126/ science.aan1121 | | Columbia-0 background with active FRI |
| Strain, strain background (*Arabidopsis thaliana*, Columbia-0) | *lhp1-3 FRI*[sf2] FLC-Venus | *Yang et al., 2017* DOI: 10.1126/ science.aan1121 | | *lhp1* mutant in Columbia-0 background with active FRI, a mutated *FLC* and complementing FLC-Venus transgene |
| Antibody | anti-H3 (rabbit polyclonal) | Abcam | Cat# ab1791, RRID:AB_302613 | '3 µg/IP' |
| Antibody | anti-H3K27me3 (rabbit, polyclonal) | Millipore | Cat# 07–449, RRID:AB_310624 | '4 µg/IP' |
| Sequence-based reagent | −86 | This study | MN-ChIP primers | F: CACTCTCGTTTACCCCCAAA R: TCCTTTTCTCGCTTTATTTCTTTC |
| Sequence-based reagent | −49 | This study and *Yang et al., 2017* DOI: 10.1126/ science.aan1121 FLC_−49_F | MN-ChIP primers | F: GCCCGACGAAGAAAAGTAG R: TTCAAGTCGCCGGAGATACT |
| Sequence-based reagent | 67 | This study | MN-ChIP primers | F: AGGATCAAATTAGGGCACAAA R: TCAATTCGCTTGATTTCTAGTTTTTT |
| Sequence-based reagent | 98 | This study | MN-ChIP primers | F: GAGAGAAGCCATGGGAAGAA R: AGCTGACGAGCTTTCTCTCGAT |
| Sequence-based reagent | 117 | This study | MN-ChIP primers | F: AAAAAACTAGAAATCAAGCGAATTGA R: CTTTCTCGATGAGACCGTT |
| Sequence-based reagent | 348 | This study | MN-ChIP primers | F: GTGCTCTTTTACTTTTCTGAG R: AGAGATCCGCCGGAAAAA |
| Sequence-based reagent | 415 | This study and *Yang et al., 2017* DOI:10.1126/ science.aan1121 FLC_416_F | MN-ChIP primers | F: GGCGGATCTCTTGTTGTTTC R: TTCTTCACGACATTGTTCTTCC |
| Sequence-based reagent | 581 | This study | MN-ChIP primers | F: TGCATGGATTTCATTATTTCCT R: TCACTCAACAACATCGAGCA |
| Sequence-based reagent | 651 | *Yang et al., 2017* DOI: 10.1126/ science.aan1121 FLC_652_F FLC_809_R | MN-ChIP primers | F: CGTGCTCGATGTTGTTGAGT R: TCCCGTAAGTGCATTGCATA |
| Sequence-based reagent | 785 | This study | MN-ChIP primers | F: TCATTGGATCTCTCGGATTTG R: AGGTCCACAGCAAAGATAGGAA |
| Sequence-based reagent | 923 | This study | MN-ChIP primers | F: TTCCTATCTTTGCTGTGGACCT R: GAATCGCAATCGATAACCAGA |
| Chemical compound, drug | Protease inhibitor c ocktail, cOmplete | Sigma- Aldrich | Cat#5056489001 | |

*Continued on next page*

*Continued*

| Reagent type (species) or resource | Designation | Source or reference | Identifiers | Additional information |
|---|---|---|---|---|
| Chemical compound, drug | Micrococcal nuclease | Takara Bio | Cat#2910A | 200 U/ml |
| Software, algorithm | ImageJ | https://imagej.nih.gov/ij/ | Fiji, RRID:SCR_002285 | |
| Software, algorithm | Image analysis pipeline | This study **Yang et al., 2017** | (with modifications) | https://github.com/JIC-Image-Analysis/lhp1-analysis |
| Other | Propidium iodide stain (PI) | Sigma-Aldrich | Cat# P4864 | 2 µg/mL |

## Simulation and model details

We simulate the status of H3K27 methylation and of the protein assembly at the *FLC* locus by expanding a previous mathematical model of a generic Polycomb target gene (*Berry et al., 2017a*; *Figure 2—figure supplement 1*). The simulations are conducted similarly, using the direct Gillespie algorithm (*Gillespie, 1977*). The details of the simulations are elaborated in this section.

In the extended model of this work, in addition to the status of the histone modifications, we also simulated how many proteins are bound in an assembly in the nucleation region of the gene. We therefore simulated both histone modifications and proteins (*Table 1*). The histone modifications are further divided into two regions, whether they are located in the nucleation region (NR) or body region (B).

In the simulations, there are six possible events: (1) the addition of a methyl group to a histone H3K27, (2) the removal of a methyl group from a histone H3K27, (3) the binding of a protein to the assembly (4), the unbinding of a protein from the assembly, (5) transcription, and (6) DNA replication.

When a methyl group is added to a histone H3K27 in the simulation, an unmethylated histone transitions to monomethylated, monomethylated to dimethylated, etc. The methylation propensity $r_i^{\mathrm{me}}$ (Table 3) is divided into two parts. $b_i^{\mathrm{me}}$ describes methylation addition from methyltransferases bound to the nucleation region that can loop to interact with distant H3K27me2/me3 and then add more H3K27me to the same or nearest neighbour nucleosomes. This term also includes addition from methyltransferases that can bind to H3K27me2/me3 and add more H3K27me to the same or nearest neighbour nucleosomes. In the model of *Berry et al., 2017a*, only the second of these possibilities was considered. Here, we generalise to add the first possibility, noting that the mathematical implementation of the model remains unchanged by this reinterpretation. We also allow for 'noisy' addition of methylation without the need for pre-existing H3K27me2/me3, but at a much lower level, as in *Berry et al., 2017a*. The second component of the methylation propensity, with parameter $k_{p\text{-}me}$, is due to feedback from the protein assembly to nucleation region H3K27 methylation, which boosts H3K27 methylation addition in the nucleation region only, provided the assembly is present.

The removal of methyl groups, demethylation, is described by $r_i^{\mathrm{dem}}$. When demethylation is carried out a trimethylated histone is converted to dimethylated, dimethylated to monomethylated etc.

The binding of a protein to the assembly is described by the propensity $r^{bind}$ (Table 3). The assembly proteins could consist in part of VRN5 and, in the cold, VIN3. VIN3 levels are known to rise approximately linearly with increasing duration of cold, and are hypothesised here to be required for

**Table 1.** Definitions of the simulated system, the histones, and the protein assembly at *FLC*.

**Definitions of the simulated system**

| Type | Status |
|---|---|
| Protein assembly occupancy $k \in [1, N_p]$ | $P_k \in \{\mathrm{unbound, bound}\}$ |
| Histones $i \in [1, N]$ | $S_i \in \{\mathrm{me0, me1, me2, me3}\}$ |
| Regions of histones | $R_m = \begin{cases} \mathrm{Histones}[1, ..., N_{NR}] \text{ when } m = NR \text{ (nucleation region)} \\ \mathrm{Histones}[N_{NR}+1, ..., N] \text{ when } m = B \text{ (body region)}, N_B = N - N_{NR} \\ \mathrm{Histones}[1, ..., N] \text{ when } m = L \text{ (entire region)}, N_L = N \end{cases}$ |

assembly formation, as VIN3 mutants are defective in nucleation. However, once the assembly has formed, we expect that VRN5 and potentially other proteins can also participate in assembly dynamics, as discussed in the main text. In the warm, VIN3 levels drop very rapidly and so VRN5 and others are likely principal constituents of the assembly in the warm following cold. To simplify the model, we consider only a single composite protein with the properties of a combined VIN3/VRN5. In the cold only, the binding propensity builds up with the number of weeks of cold (representing the VIN3 contribution, $\gamma_{bind}(w)$ in *Table 2*) and saturates with a timescale $w_0$ weeks in the cold. Once formed, with the number of protein subunits above a critical threshold (equal to $v$ in the model), positive feedback arises and contributes a much larger constant binding term (potentially representing the VRN5 contribution, $k_{bind}G$ in *Table 3*, with $k_{bind}$ a constant in all conditions). In this way nucleation of the assembly is a slow process in the cold (as observed experimentally), but once formed, strong positive feedback gives effective maintenance. When the binding of a protein to the assembly occurs in the simulations, the status of one site in the protein assembly occupancy changes from unbound to bound. The size of the assembly is the number of sites with a bound protein.

The unbinding of a protein from the assembly is described by the propensity $r_k^{unbind}$, and is the same in all conditions. For unbinding of protein $k$, the appropriate assembly occupancy status changes from bound to unbound. If sufficient proteins are lost from the assembly through $r_k^{unbind}$ or DNA replication (see below), the protein number may drop below the critical threshold $v$, thereby abolishing the strong feedback. In the warm, the remaining bound proteins will then rapidly disperse due to the lack of binding and the assembly is lost. In the cold, an assembly can still potentially reform due to the background cold-induced binding potentially via VIN3.

Transcription is described by the stepwise linear function f (*Table 3*), with maximum value $f_{lim}$. Previous work has shown that full *FLC* repression can be achieved based only on nucleation region (NR) nucleosomes (*Yang et al., 2017*). However, in the warm, once the assembly has dispersed, silencing must also be capable of being maintained by methylation on non-NR nucleosomes. This must be the case, as silencing would be deficient if only NR region nucleosomes enforced silencing in the case where the small number of NR-nucleosomes were not sufficiently inherited at DNA replication. Hence, we used a combination of both NR and non-NR region nucleosomes to control silencing. Whichever of the two regions had the higher H3K27me2/me3 level, that was the level that dictated the level of *FLC* silencing achieved, as described in *Table 3*. When transcription is carried out in the simulations, the histones are exposed to demethylation and histone exchange. The histones are demethylated (as described above) with a probability of $p_{dem}$ per histone and exchanged with a probability of $p_{ex}$ per histone. When a histone is exchanged, both histones of a nucleosome are changed to unmethylated (histone $i$ and $i+1$ if $i$ is odd, histone $i$ and $i-1$ if $i$ is even).

**Table 2.** Definitions and functions used in the propensity functions in *Table 3*.

**Definitions**

| | |
|---|---|
| The Kronecker delta | $\delta_{x,y} = \begin{cases} 1, x = y \\ 0, x \neq y \end{cases}$ |
| Neighbour effect | $E_i = \sum_{j \in M_i} \left( \rho_{me2}\delta_{S_j,me2} + \delta_{S_j,me3} \right)$ |
| Neighbouring set | $M_i = \begin{cases} \{i-3, i-2, i-1, i+1, i+2\}, i\,even, \\ \{i-2, i-1, i+1, i+2, i+3\}, i\,odd \end{cases}$ |
| Fraction of histones in me3 or me2 | $P(R_m)_{\mathrm{me2/me3}} = \frac{1}{N_m}\sum_{j \in R_m}\left( \delta_{S_j,me2} + \delta_{S_j,me3} \right)$ |
| Sum of bound proteins | $Q = \sum_{j=1}^{N_p} \delta_{P_j,\mathrm{bound}}$ |
| Protein feedback | $G = \begin{cases} 1, Q \geq v, \\ 0, Q < v \end{cases}$ |
| Spontaneous protein binding | $\gamma_{bind}(w) = \frac{\gamma_{\mathrm{VIN3}} \cdot w}{\left(1 + \frac{w}{w_0}\right)}$ $w_0$ is timescale for VIN3 binding saturation (weeks) $w$ is weeks in cold |

**Table 3.** Propensity functions used in the simulations.

**Propensities**

| | |
|---|---|
| Non-assembly methylation propensity | $b_i^{\text{me}} = \beta\big(\delta_{S_i,\text{me0}}(\gamma_{\text{me0}-1} + k_{\text{me0}-1}E_i) + \delta_{S_i,\text{me1}}(\gamma_{\text{me1}-2} + k_{\text{me1}-2}E_i) + \delta_{S_i,\text{me2}}(\gamma_{\text{me2}-3} + k_{\text{me2}-3}E_i)\big)$ |
| Demethylation propensity | $r_i^{\text{dem}} = \gamma_{dem}\big(\delta_{S_i,\text{me1}} + \delta_{S_i,\text{me2}} + \delta_{S_i,\text{me3}}\big)$ |
| Transcription | $f = \begin{cases} \alpha\left(f_{max} - \frac{P(R_m)_{me2/me3}}{P_T}(f_{max} - f_{min})\right) & , P(R_m)_{me2/me3} < P_T \\ \alpha f_{min}, P(R_m)_{me2/me3} \geq P_T \end{cases}$<br><br>$m = \begin{cases} NR, P(R_{NR})_{me2/me3} > P(R_B)_{me2/me3} \\ B, P(R_{NR})_{me2/me3} \leq P(R_B)_{me2/me3} \end{cases}$<br><br>if $f > f_{lim}$ then $f = f_{lim}$ |
| Protein binding propensity | If $Q < N_P$ then $r^{bind} = \gamma_{bind}(w) + k_{bind}G$<br>If $Q = N_P$ then $r^{bind} = 0$ |
| Protein unbinding propensity | $r_k^{unbind} = \delta_{P_k,\text{bound}}\gamma_{unbind}$ |
| Total methylation propensity | $r_i^{\text{me}} = \begin{cases} b_i^{me} + k_{p-me}G\big(\delta_{S_i,\text{me0}} + \delta_{S_i,\text{me1}} + \delta_{S_i,\text{me2}}\big), & i \in R_{NR} \\ b_i^{me}, & i \in R_B \end{cases}$ |

When replication is carried out, with a probability of 0.5 per nucleosome, the nucleosome is exchanged to a nucleosome with both histones unmethylated. In addition, with probability 0.5 for each site with a bound protein in the assembly, the assembly occupancy status is changed from bound to unbound. Thus, each individual nucleosome, and separately each individual protein in the assembly, is inherited through DNA replication with 50% probability. DNA replication is carried out when the next event in the Gillespie simulation is drawn to occur at an elapsed time since the last DNA replication equal to, or greater than, the cell cycle duration. This duration differs between cold and warm conditions (see *Tables 4* and *5*).

In *Figure 1—figure supplement 2B*, replication is simulated with an ordered, equal distribution of the nucleosomes to the daughter strands. After replication every second nucleosome is replaced with a nucleosome with both histones unmethylated. Whether odd or even numbered nucleosomes are replaced is chosen randomly with a probability of 0.5. In *Figure 1—figure supplement 2C*, ordered nucleosome distribution is combined with the assembly model described above.

The propensities are described in detail in *Tables 2* and *3*. Further justifications of the equations are described in *Berry et al., 2017a*. Note that with the exception of the assembly dynamics, and how the assembly affects H3K27me addition in the nucleation region, together with the control of silencing by both NR and non-NR region nucleosomes, the underlying model is identical to that in *Berry et al., 2017a*, although with altered parameter values (see next section).

## Model parameters and initial conditions

Some parameters of the model vary between cold and pre-/post-cold conditions; these variations are specified in *Table 4*. The cell cycle length is different between cold and warm, consistent with the observed decrease in growth when plants are in vernalization conditions (*Zhao et al., 2020*).

**Table 4.** Specification of parameters for different conditions.

Post-cold, a time lag of 1 day without replication is first simulated with new, post-cold values of the parameters specified, with the exception of $\alpha$, which is changed to its post-cold value after that time lag.

| | Pre-cold | Cold | Post-cold |
|---|---|---|---|
| Cell cycle length | $c_w$ | $c_c$ | $c_w$ |
| Trans-acting gene activation ($\alpha$) | $\alpha_w$ | $\alpha_c$ | $\alpha_w$ |
| PRC2-mediated methylation rate ($k_{me}$) at $i$-th histone | $k_{me} = \begin{cases} k_{me}^{high}, & i \in R_{NR} \\ k_{me}^{high}, & i \in R_B \end{cases}$ | $\begin{cases} k_{me}^{high}, & i \in R_{NR} \\ k_{me}^{low}, & i \in R_B \end{cases}$ | $\begin{cases} k_{me}^{high}, & i \in R_{NR} \\ k_{me}^{high}, & i \in R_B \end{cases}$ |
| Spontaneous protein binding | 0 | $\gamma_{bind}(w)$ | 0 |
| Simulated number of cell cycles $N_{gen}$ | 10.9 | $N_{gen}^x$ | $N_{gen}^y$ |

**Table 5.** Fixed parameters with references justifying values used.

| Parameter | Description | Value | Justification |
|---|---|---|---|
| $N$ | Number of histones | 60 | *Berry et al., 2017a* |
| $k_{me0-1}$ | PRC2-mediated methylation rate (me0 to me1) (histone$^{-1}$ s$^{-1}$) | $9k_{me}$ | |
| $k_{me1-2}$ | PRC2-mediated methylation rate (me1 to me2) (histone$^{-1}$ s$^{-1}$) | $6k_{me}$ | |
| $k_{me2-3}$ | PRC2-mediated methylation rate (me2 to me3) (histone$^{-1}$ s$^{-1}$) | $k_{me}$ | |
| $\gamma_{me2-3}$ | Noisy methylation rate (me2 to me3) (histone$^{-1}$ s$^{-1}$) | $k_{me2-3}/20$ | |
| $\gamma_{me1-2}$ | Noisy methylation rate (me1 to me2) (histone$^{-1}$ s$^{-1}$) | $k_{me1-2}/20$ | |
| $\gamma_{me0-1}$ | Noisy methylation rate (me0 to me1) (histone$^{-1}$ s$^{-1}$) | $k_{me0-1}/20$ | |
| $\beta$ | Relative local PRC2-activity | 1 | |
| $\rho_{me2}$ | Relative activation of PRC2 by H3K27me2 | 0.1 | |
| $\gamma_{dem}$ | Noisy demethylation rate (histone$^{-1}$ s$^{-1}$) | $f_{min}p_{dem}$ | |
| $f_{lim}$ | Limit on maximum transcription initiation (s$^{-1}$) | 1/60 | |
| $P_T$ | Threshold for full repression of transcription | 1/3 | *Hepworth et al., 2018*; *Jadhav et al., 2020* |
| $f_{min}$ | Minimum transcription initiation rate (s$^{-1}$) | $f_{max}/25$ | *Yang et al., 2014* |
| $f_{max}$ | Maximum transcription initiation rate (s$^{-1}$) | $7.5 \cdot 10^{-4}$ | *Yang et al., 2014*; *Ietswaart et al., 2017* |
| $p_{dem}$ | Demethylation probability (histone$^{-1}$ transcription$^{-1}$) | 0.17 | *Figure 2—figure supplement 2* |
| $p_{ex}$ | Histone exchange probability (histone$^{-1}$ transcription$^{-1}$) | $8.3 \cdot 10^{-2}$ | |
| $c_w$ | Cell cycle length in warm (22°C) conditions (h) | 22 | *Rahni and Birnbaum, 2019*; *Zhao et al., 2020* |
| $c_c$ | Cell cycle length in cold (5°C) conditions (h) | 154 | |

The strength of trans-activation ($\alpha$) is also changed between the pre-cold/post-cold and cold, where, post-cold, $\alpha$ is changed after a time lag of 1 day. The change in $\alpha$ in the cold is due to an observed decrease in transcription in the cold that is not caused by an increase in H3K27 methylation (*Hepworth et al., 2018*). This effect is believed to be mediated by an antisense transcription mechanism which we do not model in detail here. Instead we simply incorporate its outcome, a reduction in the push to transcribe in the cold.

The methyltransferases CLF and/or SWN are always present in the nucleation region in all conditions (*Yang et al., 2017*). Therefore, the methylation rate $k_{me}$ is set to a fixed value both in the NR and outside at all times, with one exception. The exception is a reduced rate $k_{me}^{low}$ in the cold outside of the NR. This is required to supress premature spreading of H3K27me2/me3 away from the NR in the cold. The origin for this suppression is not yet mechanistically clear. However, in our interpretation, the NR can loop over to non-NR nucleosomes and catalyse the spread of H3K27me2/me3 from a non-NR nucleosome with H3K27me2/me3 to the same nucleosome or its nearest neighbours. Either the looping interaction itself, or the strength of the catalytic reactions, may be reduced by cold temperatures.

The model parameters that are fixed throughout for all simulations are listed in *Table 5*, with references for where they are justified. Most parameter values are found in *Berry et al., 2017a*; however, a few of them are changed to adapt to the features observed at *FLC*. The parameters $f_{min}$ and $f_{max}$ are redefined (*Table 5*), and therefore new values of $p_{dem}$ and $p_{ex}$ used to ensure maintenance of low H3K27me3 levels pre-cold, but high H3K27me3 levels post-cold (*Figure 2—figure supplement 2*).

Free parameters are listed in *Table 6*. If $N_{NR}$, $N_p$, $N_{gen}^x$ and $N_{gen}^y$ are not specified, the values in *Table 6* are used. The values of $k_{me}$ were chosen in order to ensure no spreading to the body region

**Table 6.** Free parameters and their values used in the simulations.

| Parameter | Description | Value |
|---|---|---|
| $N_{\mathrm{NR}}$ | Number of histones in nucleation region | 6 |
| $N_{\mathrm{p}}$ | Maximum number of proteins in assembly | 17 |
| $N_{gen}^{x}$ <br> $N_{gen}^{y}$ | Number of cell cycles in cold conditions <br> Number of cell cycles post cold | 8.7 <br> 43.6 |
| $k_{\mathrm{me}}$ <br> $k_{\mathrm{me}}^{high}$ <br> $k_{\mathrm{me}}^{low}$ | PRC2-mediated methylation rate (me2 to me3) <br> High rate (histone$^{-1}$ s$^{-1}$) <br> Low rate (histone$^{-1}$ s$^{-1}$) | $4 \cdot 10^{-5}$ <br> $1.7 \cdot 10^{-5}$ |
| $k_{\mathrm{p-me}}$ | Assembly-mediated methylation rate (histone$^{-1}$ s$^{-1}$) | 0.01 |
| $\gamma_{VIN3}$ | Noisy addition of protein to the assembly (s$^{-1}$) | $1.8 \cdot 10^{-4}$ |
| $w_0$ | Timescale for VIN3 binding saturation in weeks | 3 |
| $k_{bind}$ | Protein feedback rate (s$^{-1}$) | 0.05 |
| $\gamma_{\mathrm{unbind}}$ | Noisy unbinding of proteins from the assembly (protein$^{-1}$ s$^{-1}$) | $10^{-3}$ |
| $v$ | Number of proteins in assembly required for protein feedback | 4 |
| $\alpha = \alpha_w$ <br> $\alpha = \alpha_c$ | Trans-acting gene activation warm conditions <br> Trans-acting gene activation cold conditions | 1 <br> 0.75 |

in the cold, but to efficiently spread post-cold. The value of $N_{NR}$ was chosen to fit the width of the nucleation region (*Figure 1B*; *Figure 1—figure supplement 1*), while $N_P$ and the other parameters were chosen to fit the rise of the nucleation peak in the cold, as well as the stability of the nucleation peak/spread state after the cold in the *lhp1* mutant and wild-type, respectively. We also varied our fitted parameters by 10% and found that our conclusions were all qualitatively unchanged by these variations.

The definition and simulations used to find the bistability measure $Bi$ (*Figure 2—figure supplement 2*) are the same as in *Berry et al., 2017a*. $Bi$ is calculated from the combined output from the simulations of a system of 60 histones initialised in each of the uniform H3K27me0 or H3K27me3 states and simulated for 50 cell cycles. $P_{OFF}$ is the proportion of the time where the system is in a high H3K27me2/me3 state and $P_{ON}$ is the proportion of time it is in a high H3K27me0/me1 state (definitions in *Table 7*). The value of $Bi$ is averaged over 4000 simulations for each parameter set. A value of $Bi$ close to 1 is regarded as a bistable simulation. The simulations are without the protein assembly to ensure the histone feedbacks alone are able to maintain the spread state after the assembly has disappeared.

The system is initialised with no proteins and all histones in me0. Before cold, the system was equilibrated for 10 days. This duration was sufficiently long to allow the system after 10 days to generate methylation and protein levels as a function of time from the previous DNA replication event, that were invariant from one cell cycle to the next. To match our experiments, the value of $N_{gen}^{x}$ was

**Table 7.** Additional parameters and equations.
$P_r$ in the definitions of $P_{OFF}$ and $P_{ON}$ refer to the proportion of time (see 'Model parameters' section), relevant for *Figure 2—figure supplement 2*.

| Parameter | Description | Value |
|---|---|---|
| $N_{sim}$ | Number of simulations for a gene | 4000 |
| $t_{out}$ | Time step for time course averaging (h) | 1 |
| | **Definition** | **Reference** |
| $P_{OFF}$ | $P_r\left(P(R_L)_{\mathrm{me2/me3}} > \frac{3P_T}{4}\right)$ | *Berry et al., 2017a* |
| $P_{ON}$ | $P_r\left(P(R_L)_{\mathrm{me2/me3}} < \frac{P_T}{4}\right)$ | |
| $Bi$ | $Bi = 4P_{ON}P_{OFF}$ | *Sneppen and Dodd, 2012*; *Berry et al., 2017a* |

chosen to match a total of 8 weeks cold exposure, with $N_{gen}^y$ representing 20 further days in the warm.

When the *lhp1* mutant is simulated starting at the end of the cold, the nucleation region is initialised with all histones with me3, and the body region with all histones me0 (*Figure 1C*, *Figure 2D*, *Figure 1—figure supplements 2* and *3*). When the protein assembly is included (*Figure 2D*, *Figure 1—figure supplement 2C*), the simulations are additionally initialised with the maximum possible number of proteins bound in the assembly. In these cases, the noisy methylation transition rates are also reduced (see Table 9) to prevent noisy addition of methylation from causing silencing, as would otherwise be possible due to the small size of the nucleation region. For these reactivation simulations, we simulated two independent copies of *FLC* for a cell, and assumed that reactivation of only one of the two copies would be sufficient for overall cellular reactivation, with reactivation (FLC-ON) occurring if the fraction of H3K27me2/me3 dropped below the fully-silenced threshold.

In *Figure 4*, PRE excision and the Lov-1 accession are simulated. For the PRE excision, the simulations are mostly the same as earlier; however, there are changes after cold treatment: for all remaining time after 12 days (including the 1 day without replication, see below), the nucleation region is changed to unmethylated, the number of proteins bound to the assembly is set to zero, $k_{me}$ in the nucleation region is set to zero and the $k_{me}$ in the body region is decreased to $k_{me}^{ex}$, as specified in *Figure 4A*. This is summarised in *Table 8*. For the simulations of Lov-1, the parameters that are different from Col*FRI* are specified in *Table 9*.

## Simulation output

When we show the simulated H3K27me3 levels and fraction of proteins in the assembly, we average the time course over $N_{sim}$ simulations (*Table 7*). In the Gillespie algorithm, the time steps are not constant. We therefore convert the time steps and the corresponding H3K27me3/protein assembly levels to constant time steps. We do so by choosing a short time step $t_{out}$ and assign it to the H3K27me3 levels that was recorded at the end of that timestep. To avoid having all the simulations synchronised, so that, for example, DNA replication does not occur at the same time for all simulations, each individual simulation is started at a random time in the cell cycle. When the simulation shifts from warm to cold conditions, we convert where the system is in the cell cycle to where it would be in the cold cell cycle, that is if there is 2 hr left before replication in the warm cell cycle, it is then 14 hr left before replication in the cold cell cycle. An analogous rule is implemented upon return to warm conditions, but only after a time lag of 1 day without replication is first simulated, to implement a delay while the plant adjusts to warm temperatures. Then, the cell cycle is converted to the warm cell cycle, that is if there is 14 hr left before replication in the cold cell cycle, it is then 2 hr left before replication in the warm cell cycle. During the 1-day lag, all parameters adopt their warm values (*Table 4*) except for the transcription activity parameter α which retains its cold value, to allow sufficient post-cold H3K27me2/me3 spreading.

For the *lhp1* mutant simulations, the one-day time lag without replication is also implemented, as above, with the reduced value of α, after which each simulated locus enters the standard warm cell cycle at a random phase. An additional 1-day lag for protein to become visible is also included.

The source code of the model simulations is available at https://github.com/ceclov/hybrid-PRC2-model copy archived at swh:1:rev:953304772e7cec326736b0c72693ba698bb2540a (*Lövkvist, 2016*).

**Table 8.** Additional details for simulations of PRE excision in *Figure 4A*.
Values for $k_{me}^{ex}$ specified in figure.

| Default $k_{me}$ | Change in simulation | Condition |
|---|---|---|
| $k_{me} = \begin{cases} k_{me}^{high}, & i \in R_{NR} \\ k_{me}^{high}, & i \in R_B \end{cases}$ | $k_{me} = \begin{cases} 0, & i \in R_{NR} \\ k_{me}^{ex}, & i \in R_B \end{cases}$ <br> $S_{NR} \in \{me0\}$ <br> $P_k \in \{\text{unbound}\}$ | After 11 days in warm, post-cold. |

**Table 9.** Summary of other figure specific parameters.

| Figure | Parameter | Value |
|---|---|---|
| 1C, 2D, *Figure 1—figure supplements 2* and *3* | FLC-ON cell | $P(R_{NR})_{\text{me2/me3}} < P_T$ |
| | $\gamma_{\text{me2-3}}$ | $k_{\text{me2-3}}/100$ |
| | $\gamma_{\text{me1-2}}$ | $k_{\text{me1-2}}/100$ |
| | $\gamma_{\text{me0-1}}$ | $k_{\text{me0-1}}/100$ |
| 1C, *Figure 1—figure supplement 3* | $\gamma_{VIN3}$ | $0$ (s$^{-1}$) |
| | $P_T$ | $1/6$ |
| | $k_{me}$, for histone $i \in R_B$ | $0$ (histone$^{-1}$ s$^{-1}$) |
| *Figure 1—figure supplement 2A* | $\gamma_{VIN3}$ | $0$ (s$^{-1}$) |
| | $P_T$ | $1/3$ |
| | $k_{me}$, for histone $i \in R_B$ | $0$ (histone$^{-1}$ s$^{-1}$) |
| *Figure 1—figure supplement 2B* | $\gamma_{VIN3}$ | $0$ (s$^{-1}$) |
| | $P_T$ | $1/6$ |
| | $k_{me}$, for histone $i \in R_B$ | $0$ (histone$^{-1}$ s$^{-1}$) |
| | Regular nucleosome distribution to daughter strands | N/A |
| *Figure 1—figure supplement 2C* | $\gamma_{VIN3}$ | $0$ (s$^{-1}$) |
| | $P_T$ | $1/3$ |
| | $k_{me}$, for histone $i \in R_B$ | $0$ (histone$^{-1}$ s$^{-1}$) |
| | Regular nucleosome distribution to daughter strands | N/A |
| | $\gamma_{\text{unbind}}$ | $8 \cdot 10^{-4}$(protein$^{-1}$ s$^{-1}$) |
| 2C | $P_T$ | $1/3$ |
| | $k_{me}$, for histone $i \in R_B$ | $1.7 \cdot 10^{-5}$ (histone$^{-1}$ s$^{-1}$) |
| 2D | $\gamma_{VIN3}$ | $0$ (s$^{-1}$) |
| | $P_T$ | $1/3$ |
| | $k_{me}$, for histone $i \in R_B$ | $0$ (histone$^{-1}$ s$^{-1}$) |
| 4 B,C | $\gamma_{VIN3}$ | $2.1 \cdot 10^{-4}$(s$^{-1}$) |
| | $N_{gen}^x$ | $13.1$ |
| | $\alpha_w$ | $2$ |
| | $k_{me}^{low}$ | $1.7 \cdot 10^{-5}$(histone$^{-1}$ s$^{-1}$) |
| *Figure 4—figure supplement 1* | As 4B,C but no replication in warm after cold | N/A |

## Processing of published experimental data

The ChIP data in *Figures 2C* and *3A* are from *Yang et al., 2014*. The data in *Figure 4B* are from *Qüesta et al., 2020*. To measure H3K27me3 levels for the nucleation region and the body region separately, primers for those specific regions were used (*Table 10*) and the average value separately calculated over each region. The sequences and positions of these primers are specified in the corresponding Supplementary Information of the above references. For the nucleation region and body regions separately, the lowest value before or during the cold is then subtracted from each time point. The resulting values for the nucleation and body regions are then normalised to the value of the nucleation region at the longest cold treatment (8 weeks in Col*FRI*, 12 weeks in Lov-1). This procedure accounts for loci in our experiments that have nucleated/spread before cold treatment and allows us to focus on loci that are un-nucleated at the start of cold, as appropriate for our simulations.

**Table 10.** Primers used to define nucleation and body region in ChIP data for Lov-1 (*Qüesta et al., 2020*) and Col*FRI* (*Yang et al., 2014*).

| | Lov-1 | Col-*FRI* |
|---|---|---|
| Nucleation region | 157_F 314_R | 157_F 314_R |
| | 416_F 502_R | 416_F 502_R |
| Body | 2465_F 2560_R | 1933_F 2171_R |
| | 3197_F 3333_R | 2465_F 2560_R |
| | 4322_F 4469_R | 3197_F 3333_R |
| | 5139_F 5244_R | 3998_F 4178_R |
| | | 4322_F 4469_R |
| | | 5139_F 5244_R |

Note that we only use ChIP data during but not after the cold to constrain the models. This is because the increase in H3K27me3 levels in the warm seen in the body region requires an active cell cycle (*Yang et al., 2017*). Since only a subset of the cells in the plant are actively cycling and dividing, fitting the rise in body H3K27me3 requires knowledge not only of the intrinsic spreading process at a single locus, but also of the relative fraction of cells that are dividing and therefore capable of undergoing spreading. Due to this additional complexity, which cannot yet be fully controlled, we only fit to ChIP data in the cold, where there is less division and so these complications are reduced. Note that this complication does not apply to the root imaging data, since all the cells being imaged are actively cycling and dividing.

Experimental data in *Figure 1C*, *Figure 2D*, *Figure 1—figure supplements 2* and *3* were processed as described in *Yang et al., 2017* and as described below. *VIN3* expression levels in *Figure 2B* are from Col*FRI* with 8°C cold treatment (*Hepworth et al., 2018*).

## Experimental details

In this section, we describe experimental details of the imaging and quantification of *FLC* expression in *lhp1* (*Figure 1C*, *Figure 2D*, *Figure 1—figure supplements 2* and *3*), as well as for the ChIP and MNase experiments (*Figure 1B* and *Figure 1—figure supplement 1*).

## Plant material

When *lhp1* is used it refers to *lhp1-3* which is described in *Larsson et al., 1998*. To image FLC levels in *lhp1*, a transgenic line with *FLC-Venus* in the *FRI^{sf2}* background (described in *Yang et al., 2017*) was used. For the MN-ChIP experiment, Col*FRI* was used.

## Growth conditions

For both experiments, the seeds were vapour-phase sterilised as described in *Clough and Bent, 1998* and sown on Murashige and Skoog (MS) medium plates without glucose (see *Table 11* for individual experiments). The plates were kept at 4°C in the dark for 2 days for stratification. Before vernalization, or for non-vernalized conditions (NV), the plants were grown in long-day conditions (16 hr light, 8 hr darkness with constant 20°C). When plants were vernalized, they were first pre-grown in NV conditions (number of days specified in *Table 11*) and then transferred to vernalization conditions (8 hr light, 16 hr darkness with constant 5°C). For the *lhp1*-imaging, the plants were further transferred back to long-day conditions.

## *lhp1* imaging

The roots were imaged on microscope slides (VWR, 631–0117) with No. 1,5 coverslips (VWR, 631–0138). The slips were sealed using a frame of adhesive tape (0.12 mm thickness, Sigma-Aldrich, GBL620001-1EA). The coverslip was pressed on the adhesive tape to prevent drying during imaging. The primary roots were chosen and incubated on the slide in 30 μL 1xPBS containing 2 μg/mL propidium iodide (PI, Sigma-Aldrich, P4864) and incubated for 1 min before imaging. The imaging was performed on a Leica SP8 X confocal microscope using a 20X/0.75NA objective lens with water immersion, with illumination at 514 nm (Argon ion laser, 5% intensity). For the Venus channel, emissions were detected between wavelengths 518 nm and 550 nm by a cooled detector (Leica HyD SMD) in photon-counting mode. Propidium Iodide (PI) was detected simultaneously with FLC-Venus, by collecting emissions between wavelengths 610 nm and 680 nm in standard imaging mode. For each root, a z-step size of 0.95 μm was used, starting from the upper surface and then 22–40 z-slices

**Table 11.** Summary of experimental details for *lhp1* imaging and MN-ChIP.

|  | *lhp1* imaging | MN-ChIP |
| --- | --- | --- |
| Plant material | *FRI lhp1-3 FLC-Venus* | Col*FRI* |
| Growth media | MS-GLU (MS without glucose) | MS-GLU (MS without glucose) |
| Plates | Vertical | Horizontal |
| Pre-growth | 7 days | 14 days |

depending on the thickness of the root. An image format of 1024 x 512 pixels with a 1.75X zoom factor was used.

The images were analysed to generate mean FLC-Venus intensities per cell using a processing pipeline, previously used in *Yang et al., 2017* and available at https://github.com/JIC-Image-Analysis/root-3d-segmentation (*Hartley and Olsson, 2016*). The mean FLC-Venus intensity for a cell was determined by dividing the total FLC intensity by the total cell volume (in voxels). For the last time point, 14 days after a 10-week cold treatment, the pipeline was adapted to allow precise parameter tuning of the segmentation to correct for difficulties in imaging. This new pipeline is available at https://github.com/JIC-Image-Analysis/lhp1-analysis (*Hartley, 2020*). In this pipeline, segmentations, PI cell wall information and FLC-Venus fluorescence are combined to generate mean intensity measurements. Segmentations are loaded and reconstructed cells touching the border of the image are removed. The remaining cells are then rank ordered by size, and the top 300 cells (determined by examination of numbers of reconstructed cells in earlier time points) used to measure FLC-Venus intensity. Mean FLC-Venus intensity was then calculated for each of these remaining reconstructed cells by dividing total FLC intensity by the cell volume in voxels.

For each timepoint except the last at 14 days post cold, the background levels were estimated as at or below the 98-th percentile of mean cellular FLC-Venus intensity in roots at that timepoint where all cells exhibited silenced FLC-Venus. Cells above this threshold were defined as FLC-ON, and the proportion of FLC-ON cells was calculated for each time point (*Figure 1C*, *Figure 2D*, *Figure 1—figure supplements 2* and *3*). For the last time point, 14 days after 10 weeks of cold, all roots had some FLC-ON cells and therefore the 85-th percentile was used to conservatively define the FLC-ON cells. In *Figure 1—figure supplement 3*, we compare the histone-feedback only model to this data, concluding that even when using the 40-th percentile as a cut-off, it is still not possible to satisfactorily fit the model. Note that to ensure that the argon laser of Leica SP8X was stable after turning on, the five first roots imaged at each time-point were excluded. The statistics of the cells included in *Figure 1—figure supplement 3* are summarised in *Table 12*.

## Interpreting FLC-Venus fluorescence levels at early time points after cold

*FLC* expression is low after 6 weeks of cold treatment in *lhp1* and *FRI^{sf2}* (*Yang et al., 2017*). However, in *Berry et al., 2015*, it was shown that even though there is FLC-Venus signal post 6 weeks vernalization, there is very little FLC-Venus protein present after 6 weeks cold in *FRI^{sf2}* in an immunoblot. Here, we therefore treated the plants with 10 weeks of cold, to fully silence *FLC*. Some Venus intensity is nevertheless still observed 1, 2, and 4 days following return to warm after 10 weeks of cold, particularly at the 1-day time point. Furthermore, the Venus signal observed at these early time points, 1, 2, and 4 days after cold, is not located in the nucleus (*Figure 1—figure supplement 4*, top). This finding indicates that the early accumulation is a cold-induced phenomenon that might be Venus-specific. For these reasons, we therefore exclude these time points from our analysis, as they do not reflect transcriptional reactivation of *FLC* in the roots. In contrast, the Venus signal at later timepoints (7, 10, and 14 days after cold) is nuclear-localised (*Figure 1—figure supplement 4*, bottom), indicating that data from these time points do represent genuine *FLC* reactivation.

The images in *Figure 1—figure supplement 4* were processed with a Gaussian blur (Fiji, 0.2 μm filter size) to the FLC-Venus channel. A sum projection over 9 z-slices (8.55 μm) was performed. For the cell wall stain, the central z-plane was extracted and overlaid on the FLC-Venus sum projection.

**Table 12.** Summary statistics for quantitative image analysis of *lhp1*.

| Days after cold treatment | Number of roots imaged | Number of cells quantified |
| --- | --- | --- |
| T7 | 11 | 1696 |
| T10 | 17 | 3598 |
| T14 | 10 | 2648 |

## MN-ChIP experiments

MN-ChIP experiments were performed as described in *Yang et al., 2017* with modifications and omitting a strong sonication step. After washes with Honda buffer, the nuclei were washed once with Resuspension buffer (10 mM Tris-HCl pH=8; 5 mM NaCl; 2.5 mM CaCl2; 2 mM MgCl2; Protease inhibitor cocktail (Sigma, #5056489001)). Next, the pellet was weighted and resuspended in Resuspension buffer in a 1:5 ratio (mg pellet: µL buffer). 150 µL nuclei were added to commercial 10x Takara MNase buffer (Takara Bio, #2910A) and Micrococcal nuclease (Takara Bio, #2910A) to obtain 200 U/ml enzyme concentration and incubated at 37°C for 45 min. Suitable enzyme concentrations were determined experimentally to obtain chromatin fragmented to mononucleosomal resolution. The reaction was stopped by the addition of 25 mM EGTA (final concentration). Afterwards, the samples were sonicated twice using Bioraptor Pico (Diagenode, #B01060010) in mild conditions: 10 s ON/50 s OFF (low settings) and used for immunoprecipitation as described in *Yang et al., 2017*. The antibodies were: anti-H3 (Abcam, #ab1791) and anti-H3K27me3 (Millipore, #07–44).

## Acknowledgements

We thank Rea Antoniou-Kourounioti, Geng-Jen Jang, Silvia Costa and Govind Menon for critical reading of the manuscript, and all members of the Howard and Dean groups for discussions.

## Additional information

### Funding

| Funder | Grant reference number | Author |
| --- | --- | --- |
| Biotechnology and Biological Sciences Research Council | GEN (BB/P013511/1) | Caroline Dean<br>Martin Howard |
| UK Research and Innovation | EP/T00214X/1 | Caroline Dean<br>Martin Howard |

The funders had no role in study design, data collection and interpretation, or the decision to submit the work for publication.

### Author contributions

Cecilia Lövkvist, Conceptualization, Data curation, Software, Formal analysis, Validation, Investigation, Visualization, Methodology, Writing - original draft, Project administration, Writing - review and editing; Pawel Mikulski, Svenja Reeck, Data curation, Methodology, Writing - review and editing; Matthew Hartley, Data curation, Software, Methodology, Writing - review and editing; Caroline Dean, Resources, Supervision, Funding acquisition, Writing - review and editing; Martin Howard, Conceptualization, Resources, Supervision, Funding acquisition, Validation, Writing - original draft, Project administration, Writing - review and editing

### Author ORCIDs

Cecilia Lövkvist ⓘD https://orcid.org/0000-0001-7696-7814
Caroline Dean ⓘD http://orcid.org/0000-0002-6555-3525
Martin Howard ⓘD https://orcid.org/0000-0001-7670-0781

### Decision letter and Author response

Decision letter https://doi.org/10.7554/eLife.66454.sa1
Author response https://doi.org/10.7554/eLife.66454.sa2

## Additional files

### Supplementary files

- Transparent reporting form

## Data availability

The raw images of roots that were used to extract FLC reactivation dynamics are available at doi:10.5061/dryad.866t1g1rd. Source data files have been provided for the MN-ChIP in Figure 1B and Figure 1 - supplement 1 and for the fraction of FLC-ON cells used in Figure 1C, 2D and Figure 1 - supplement 2.

The following dataset was generated:

| Author(s) | Year | Dataset title | Dataset URL | Database and Identifier |
|---|---|---|---|---|
| Reeck S, Lövkvist C, Mikulski P, Hartley M, Dean C, Howard M | 2021 | lhp1 FLC-Venus time course imaging – Hybrid protein assembly-histone modification mechanism for PRC2-based epigenetic switching and memory | https://10.5061/dryad.866t1g1rd | Dryad Digital Repository , 10.5061/dryad.866t1g1rd |

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
