## [Decision Letter]

**Acceptance summary:**

The paper tackles a central question in epigenetics, namely how chromatin-based gene regulatory information can be propagated through cell cycles. They show that incoorporating additional memory elements into a mathematical model, in the form of localized, heritable, protein oligomers recapitulates experimental data.

**Decision letter after peer review:**

Thank you for submitting your article "Hybrid protein oligomer-histone modification mechanism for PRC2-based epigenetic switching and memory" for consideration by *eLife*. Your article has been reviewed by 2 peer reviewers, and the evaluation has been overseen by Aleksandra Walczak as the Senior and Reviewing Editor. The following individual involved in review of your submission has agreed to reveal their identity: Nicole Francis (Reviewer #1).

Essential revisions:

Please respond to Reviewer 1's comments.

*Reviewer #1:*

In this paper, Lovkvist et al. develop a model for epigenetic memory based on regulation of the FLC locus in Arabidopsis by PRC2 and H3K27 methylation. This model captures epigenetic switching and memory, and the processes of nucleation and spreading of histone modifications. The model is fundamentally different from "classical" histone read-write based models (including from this group) in introducing a second memory element, here proposed as a locally formed protein oligomer. This group previously developed a model that could explain the generation and stability of large domains of H3K27me3 (including at the FLC gene) using PRC2 read-write mechanism as the main feature. However, their observation by this team of "metastable" silencing at FLC in mutants (like lhp1) that nucleate H3K27me3 but do not spread it over the locus prompted them to re-evaluate the model. The observed metastable silencing seems to depend on only three highly methylated nucleosomes, too few to maintain silencing in the previous read-write based model. The authors therefore surmised that there must be a mechanism to allow very small chromatin domains (like the nucleation region of FLC) to propagate. They propose two options: 1) precise segregation of parental histones between strands; 2) additional memory elements (proposed as locally formed protein oligomers) at the nucleation site. The first model is implemented and rejected, (see point 1 below) since it does not capture the experimental data. The second idea, the formation of heritable protein oligomers at the nucleation site, is developed into a model that can capture the dynamics of the system and predicts ~17 "memory elements" are required to do so. This model is developed using the simplified lhp1 mutant (which only has H3K27me3 at the nucleation site and shows metastable silencing). The authors use VIN3 as a candidate protein oligomer, although direct evidence for this is lacking, and indeed VIN3 expression is consistent with this function in the early, but not later part of the process The model is then generalized to a wild-type scenario by adding a looping read-write based mechanism to spread H3K27me3. The model also responds to removing the nucleation region by losing silencing and H3K27me3 similar to the effect of deleting a PRE (Polycomb recruiting elements in *Drosophila*), further supporting its generality to Polycomb regulation in other systems. Finally, the model could be altered to capture the altered FLC regulation dynamics in an Arabidopsis genetic variant.

The key new aspects to the model are the requirement for an additional memory element to capture the observed dynamics, and the specific hypothesis that this element is in the form of heritable protein oligomers. This model may be generalizable to Polycomb regulation in other systems. The ability of the model to predict (meta)stable chromatin states that involve small chromatin domains could also have more general relevance beyond Polycomb function. The complete rejection of balanced segregation of parental histones in the model would need to be reconsidered in light of recent data. Additional explanation of the envisioned mechanism of nucleation of protein oligomers in light of what is known about Polycomb protein recruitment mechanisms would need to be provided. The model also recapitulates H3K27me3 dynamics well but loss of metastable gene silencing less well.

Comments for the authors:

1) The authors consider a model for balanced segregation of histones (Figure SX), but reject this model because it does not fit the experimental data well. However, I do not see why this model should be mutually exclusive with the protein oligomer model-I think it would be appropriate to consider a model with both the protein oligomer and balanced histone segregation. The data from both yeast and mammalian cells indicating that chaperone activities at DNA replication forks promote either leading or lagging strand deposition makes it not only possible but likely that segregation is not random (10.1126/science.aau0294, DOI: 10.1126/science.aat8849, DOI: 10.1016/j.molcel.2018.09.001) Work showing regulated biased segregation in *Drosophila* germline stem cells (from the Chen group DOI: 10.1038/s41594-019-0269-z) provides additional evidence that this process can be regulated.

2) While I think the author's hypothesis is likely to be correct, and is very exciting, I find that certain features of the model seem highly specific in cases where data ruling out other possibilities does not exist. I don't think the model needs to be changed, but I think acknowledging other possibilities makes the ideas behind the model more general.

– page 7 lines 252-253 why do the protein oligomers segregate randomly? it seems equally possible that they segregate equally, or may not segregate at all but be shared between replicated chromatids prior to sister resolution.

– page 7 line 262-264 is it essential that the new memory elements are required to initially create the nucleation site? couldn't there be a separate mechanism for this?

– page 7 lines 290-292-why is it assumed that the oligomers activate rather than recruit PRC2 (i.e. increase its binding at the nucleation site)

– page 8 line 323-324 "we note that the oligomer size before replication must be more than twice the minimal size needed for a stable oligomer…." can the authors explicitly relate this to the conclusion that ~17 memory elements are needed-this is before replication?

– page 9 lines 349-351 Please explain why VIN3 is specifically found at the nucleation site (e.g. is it recruited by something else?)

– page 11 line 440-441 "the oligomer's only effect on histones is to enhance H3K27me3 in the nucleation region" – Why would this be the case, and is it essential? If looping involves contact between PRC2 at the nucleation site and the distal nucleosomes, it seems quite likely that the effects of the protein oligomer would still occur (unless the oligomer recruits rather than stimulates PRC2).

3) In figure 1C, the "peak" of H3K27me3 is not obvious. From both these data and the supplemental data, it seems like H3K27me3 (when normalized to H3) is similar across the region; without a true negative for comparison, I do not see how these data indicate a 3-nucleosome wide peak (which is fundamental in the model). The published data from Yang et al. (Science 2017) is more obvious.

4) It looks as though the model recapitulates H3K27me3 dynamics at the nucleation region well (Figure 2C), but loss of metastable silencing less well (Figure 2D) (although additional data points would be very helpful here). Do the authors have an explanation for this?

5) The authors invoke a looping mechanism to explain spreading of H3K27me3, but it would be interesting to consider the model of Chory et al. doi: 10.1016/j.molcel.2018.10.028, which also can explain spreading from a nucleation site, but through a distinct mechanism (histone exchange).

6) It would also be interesting to consider (perhaps in the discussion) the model of Pease et al. (DOI:https://doi.org/10.1016/j.celrep.2021.108888), which is somewhat analogous to what is proposed here in requiring a two step mechanism to explain H3K27me3 dynamics and silencing, but invokes chromatin compaction as the "other" element.

7) The usual explanation for the requirement for silencing elements in yeast and PREs in *Drosophila* is that sequence specific DNA binding proteins that recruit Polycomb proteins continuously recognize these elements, thereby recruiting chromatin modifying enzymes. Here, it is proposed that locally formed protein oligomers serve this function. I think the relationship between recruiting TFs and the stochastic nucleation of protein oligomers could be explained more clearly. Is there still a role for recruiting TFs (or other recruitment mechanisms)? How are the protein oligomers formed at the correct genomic location? Are oligomer forming proteins recruited by TFs (particularly in the case of PREs)?

*Reviewer #2:*

This manuscript discusses PRC2 based epigenetic switching and memory at the FLC locus in *Arabidopsis thaliana*. PRC epigenetic memory is linked to H3K27me3 through a read/write maintenance mechanism, epigenetic state switching and memory over many cell cycles. To explain such a persistence, the authors introduce a mathematical model envisaging an extra protein memory storage with oligomeric feedback that persist through replication, in addition to histone modifications.

I find the paper very well written, scientifically sound and very interesting. Additionally, the proposed model describes a generic mechanism that could be widely applicable.

---

## [Author Response]

Reviewer #1:[…] Comments for the authors:1) The authors consider a model for balanced segregation of histones (Figure SX), but reject this model because it does not fit the experimental data well. However, I do not see why this model should be mutually exclusive with the protein oligomer model-I think it would be appropriate to consider a model with both the protein oligomer and balanced histone segregation. The data from both yeast and mammalian cells indicating that chaperone activities at DNA replication forks promote either leading or lagging strand deposition makes it not only possible but likely that segregation is not random (10.1126/science.aau0294, DOI: 10.1126/science.aat8849, DOI: 10.1016/j.molcel.2018.09.001) Work showing regulated biased segregation in *Drosophila* germline stem cells (from the Chen group DOI: 10.1038/s41594-019-0269-z) provides additional evidence that this process can be regulated.

Yes, thank you for raising this point and for the additional references (now included). It is indeed possible that both ordered segregation occurs and that extra memory elements exist. We now simulate a combination and show that the persistent memory in the *lhp1* mutant can be recapitulated with ordered segregation of nucleosomes together with a smaller assembly of about 8 elements. We have added a new figure panel: Figure 1 – supplement 2C to show this and extra discussion at lines 367-373.

2) While I think the author's hypothesis is likely to be correct, and is very exciting, I find that certain features of the model seem highly specific in cases where data ruling out other possibilities does not exist. I don't think the model needs to be changed, but I think acknowledging other possibilities makes the ideas behind the model more general.– page 7 lines 252-253 why do the protein oligomers segregate randomly? it seems equally possible that they segregate equally, or may not segregate at all but be shared between replicated chromatids prior to sister resolution.

This is a very good point: we chose random segregation to make the dynamics of the assembly conceptually similar to those of nucleosomes. It is nevertheless possible that the distribution might be more ordered (as we explicitly consider for the nucleosomes). However, equal segregation is not likely as this would lead to memory that is very persistent, which is not what we observe experimentally in the *lhp1* mutant. We have added some extra discussion of this important point at lines 297-299.

– page 7 line 262-264 is it essential that the new memory elements are required to initially create the nucleation site? couldn't there be a separate mechanism for this?

It is possible that there could be a separate mechanism. However, utilising the memory elements to also perform nucleation naturally solves two issues: why is nucleation so slow (low probability for sufficient monomers to come together to initiate the assembly) and what explains the persistence of the memory (assembly subunits as extra memory elements). We now elaborate on these ideas at lines 331-333.

– page 7 lines 290-292-why is it assumed that the oligomers activate rather than recruit PRC2 (i.e. increase its binding at the nucleation site)

In principle, it is also possible that the oligomers recruit PRC2. However, our prior experiments (Yang et al., Science 2017) indicate that the PRC2 methyltransferase CLF, for example, is already present in the nucleation region prior to cold. Hence, we favour activation for CLF. However, other PRC2 components could be recruited and hence a combination of recruitment and activation is perhaps most likely. In terms of our mathematical model, the model does not distinguish between these two possibilities. We have amended the text at lines 270-272 to cover this point.

– page 8 line 323-324 "we note that the oligomer size before replication must be more than twice the minimal size needed for a stable oligomer…." can the authors explicitly relate this to the conclusion that ~17 memory elements are needed-this is before replication?

Yes, we now discuss this further: the number of protein subunits needed for positive feedback is 4. Therefore, random partitioning of 17 elements before replication is highly likely to produce more than the 4 elements after replication needed for strong feedback. However, there is still some possibility of dropping below this threshold or getting sufficiently close to it that even strong feedbacks do not always rebuild the larger assembly. This is why the memory in the *lhp1* mutant is not completely persistent. We now discuss these issues at lines 306-308.

– page 9 lines 349-351 Please explain why VIN3 is specifically found at the nucleation site (e.g. is it recruited by something else?)

We have previously shown that VIN3 indirectly interacts with an upstream element (VAL1), which itself has DNA sequence specific binding (see Questa et al., Science 2016). We have made our discussion on this point more explicit at lines 278-280.

– page 11 line 440-441 "the oligomer's only effect on histones is to enhance H3K27me3 in the nucleation region" – Why would this be the case, and is it essential? If looping involves contact between PRC2 at the nucleation site and the distal nucleosomes, it seems quite likely that the effects of the protein oligomer would still occur (unless the oligomer recruits rather than stimulates PRC2).

We made this assumption based on the behaviour of H3K27me3 in the nucleation region and outside. Even after spreading we see that the nucleation region has higher levels of H3K27me3 than the outside, body region. If the assembly was boosting PRC2 activity uniformly, then we would of course expect similar levels of H3K27me3 in both nucleation and body regions. Since this is not the case, we assumed that the assembly only boosts PRC2 activity in the nucleation region. Once the assembly disappears, H3K27me3 in the nucleation region in the model returns to the same level as the body, as seen experimentally. Hence, for reasons that we do not fully understand at present, the assembly would seem to only boost PRC2 activity locally at the nucleation region and not across the whole locus. We have tried to bring out this reasoning better in lines 471-474.

3) In figure 1C, the "peak" of H3K27me3 is not obvious. From both these data and the supplemental data, it seems like H3K27me3 (when normalized to H3) is similar across the region; without a true negative for comparison, I do not see how these data indicate a 3-nucleosome wide peak (which is fundamental in the model). The published data from Yang et al. (Science 2017) is more obvious.

Apologies, we were not clear on this point. The data from Yang et al. indeed demonstrates the three nucleosome width; the new data allows us to resolve the individual nucleosomes within this peak to show that they all contribute to some degree. We have rewritten lines 159-166 to make this clear.

4) It looks as though the model recapitulates H3K27me3 dynamics at the nucleation region well (Figure 2C), but loss of metastable silencing less well (Figure 2D) (although additional data points would be very helpful here). Do the authors have an explanation for this?

It is correct that the loss of metastable silencing is fit slightly less well. There could be various reasons for this, including a more ordered distribution of nucleosomes to daughter strands. We discuss this at lines 367-373. We also emphasise that it is very difficult to gain more data points than the three we currently have: each data point encapsulates a huge amount of FLC-Venus imaging data collected from many roots. Earlier time points in the warm are confounded by artefacts in the imaging (see Figure 1 – supplement 4) and at later time points the plants become very large, with equivalent plant material to earlier timepoints difficult to collect when growing using plates.

5) The authors invoke a looping mechanism to explain spreading of H3K27me3, but it would be interesting to consider the model of Chory et al. doi: 10.1016/j.molcel.2018.10.028, which also can explain spreading from a nucleation site, but through a distinct mechanism (histone exchange).

Thanks – we now briefly mention the alternative spreading mechanism at lines 401-402.

6) It would also be interesting to consider (perhaps in the discussion) the model of Pease et al. (DOI:https://doi.org/10.1016/j.celrep.2021.108888), which is somewhat analogous to what is proposed here in requiring a two step mechanism to explain H3K27me3 dynamics and silencing, but invokes chromatin compaction as the "other" element.

The Pease et al. model is definitely interesting and could be an alternative way to maintain memory states. However, we should point out that the simulations in Pease et al. use 50-100 nucleosomes and the authors point out that “below a threshold number of nucleosomes, the chromatin assembly is unstable and dissolves, leading to locus decompaction and consequent gene expression”. In our case, the compact assembly would have to apply down to the level of only 3 nucleosomes, which seems unlikely. For this reason, we do not favour this explanation. We have added discussion on this point to lines 548-552.

7) The usual explanation for the requirement for silencing elements in yeast and PREs in *Drosophila* is that sequence specific DNA binding proteins that recruit Polycomb proteins continuously recognize these elements, thereby recruiting chromatin modifying enzymes. Here, it is proposed that locally formed protein oligomers serve this function. I think the relationship between recruiting TFs and the stochastic nucleation of protein oligomers could be explained more clearly. Is there still a role for recruiting TFs (or other recruitment mechanisms)? How are the protein oligomers formed at the correct genomic location? Are oligomer forming proteins recruited by TFs (particularly in the case of PREs)?

It is indeed possible that TFs could play the recruitment role rather than the assembly proposed here. However, we were led to the assembly hypothesis by the previously known behaviour of the VIN3/VRN5 proteins (which are not TFs). The correct genomic location for these factors is specified in part by other upstream proteins like VAL1 (see above), which is a DNA sequence specific transcriptional repressor. We now discuss these issues at lines 278-280 in the manuscript.